# YAP1 subgroup supratentorial ependymoma requires TEAD and nuclear factor I-mediated transcriptional programmes for tumorigenesis

Kristian W. Pajtler[1,2,3,17], Yiju Wei[4,17], Konstantin Okonechnikov[1,2,17], Patricia B.G. Silva[1,2,17], Mikaella Vouri[1,2], Lei Zhang [4], Sebastian Brabetz[1,2], Laura Sieber[1,2], Melissa Gulley[4], Monika Mauermann[1,2], Tatjana Wedig[1,2], Norman Mack[1,2], Yuka Imamura Kawasawa[5,6], Tanvi Sharma[1,2], Marc Zuckermann[1,2], Felipe Andreiuolo[7], Eric Holland [8], Kendra Maass[1,2], Huiqin Körkel-Qu[9], Hai-Kun Liu[9], Felix Sahm [10,11], David Capper[12], Jens Bunt [13], Linda J. Richards [13], David T.W. Jones[1,14], Andrey Korshunov[10,11], Lukas Chavez [1,2], Peter Lichter[15], Mikio Hoshino[16], Stefan M. Pfister [1,2,3], Marcel Kool[1,2], Wei Li [4,5] & Daisuke Kawauchi [1,2]

YAP1 fusion-positive supratentorial ependymomas predominantly occur in infants, but the molecular mechanisms of oncogenesis are unknown. Here we show YAP1-MAMLD1 fusions are sufficient to drive malignant transformation in mice, and the resulting tumors share histo-molecular characteristics of human ependymomas. Nuclear localization of YAP1-MAMLD1 protein is mediated by MAMLD1 and independent of YAP1-Ser127 phosphorylation. Chromatin immunoprecipitation-sequencing analyses of human YAP1-MAMLD1-positive ependymoma reveal enrichment of NFI and TEAD transcription factor binding site motifs in YAP1-bound regulatory elements, suggesting a role for these transcription factors in YAP1-MAMLD1-driven tumorigenesis. Mutation of the TEAD binding site in the YAP1 fusion or repression of NFI targets prevents tumor induction in mice. Together, these results demonstrate that the YAP1-MAMLD1 fusion functions as an oncogenic driver of ependymoma through recruitment of TEADs and NFIs, indicating a rationale for preclinical studies to block the interaction between YAP1 fusions and NFI and TEAD transcription factors.

[1] Hopp-Children's Cancer Center Heidelberg (KiTZ), 69120 Heidelberg, Germany. [2] Division of Pediatric Neurooncology, German Cancer Research Center (DKFZ), 69120 Heidelberg, Germany. [3] Department of Pediatric Hematology and Oncology, Heidelberg University Hospital, 69120 Heidelberg, Germany. [4] Division of Pediatric Hematology and Oncology, Department of Pediatrics, Penn State Health Hershey Medical Center, Penn State College of Medicine, Hershey, PA 17033, USA. [5] Department of Biochemistry and Molecular Biology, Penn State Health Hershey Medical Center, Penn State College of Medicine, Hershey, PA 17033, USA. [6] Department of Pharmacology, Penn State Health Hershey Medical Center, Penn State College of Medicine, Hershey, PA 17033, USA. [7] Department of Neuropathology, Ste. Anne Hospital, 75014 Paris, France. [8] Human Biology Division, Fred Hutchinson Cancer Research Center, Seattle, WA 98109, USA. [9] Division of Molecular Neurogenetics, German Cancer Research Center (DKFZ), 69120 Heidelberg, Germany. [10] Clinical Cooperation Unit Neuropathology, German Cancer Research Center (DKFZ), 69120 Heidelberg, Germany. [11] Department of Neuropathology, Heidelberg University Hospital, 69120 Heidelberg, Germany. [12] Charité – Universitätsmedizin Berlin, corporate member of Freie Universität Berlin, Humboldt-Universität zu Berlin, and Berlin Institute of Health, Department of Neuropathology, Partner Site Berlin, German Cancer Research Center (DKFZ), Heidelberg, Germany. [13] Queensland Brain Institute, The University of Queensland, Brisbane 4072, Australia. [14] Pediatric Glioma Research Group, German Cancer Research Center (DKFZ), 69120 Heidelberg, Germany. [15] Division of Molecular Genetics, German Cancer Research Center (DKFZ), 69120 Heidelberg, Germany. [16] Department of Biochemistry and Cellular Biology, National Institute of Neuroscience, NCNP, Tokyo, Japan. [17] These authors contributed equally: Kristian W. Pajtler, Yiju Wei, Konstantin Okonechnikov, Patricia B. G. Silva. Correspondence and requests for materials should be addressed to W.L. (email: weili@pennstatehealth.psu.edu) or to D.K. (email: d.kawauchi@kitz-heidelberg.de)

Ependymomas (EPNs) are neuroepithelial tumors that occur in all age groups, with different predominant locations along the neuraxis. In children, EPNs account for 10% of all malignant central nervous system tumors, of which 30% occur in children under 3 years of age[1–3]. EPNs are chemotherapy-resistant tumors, which mostly lack actionable molecular targets, so neurosurgical intervention plays a primary role for local tumor control[4,5]. In addition to resection, post-operative involved-field high-dose radiotherapy is considered the standard-of-care for children older than 12 months and with non-disseminated disease[6,7]. Although these approaches may effectively reduce the risk of EPN recurrence, long-term sequelae steadily increase with therapy intensification, especially in infants. Thus, for very young EPN patients, both reliable risk stratification and new treatment options are urgently needed.

Through recent collaborative efforts on molecular characterization of brain tumors, we have identified molecularly distinct groups of EPNs arising from the three anatomic compartments of the central nervous system[8]. This molecular classification outperforms the current histopathological classification regarding clinical associations[8]. Within the supratentorial (ST) compartment, two molecular groups, ST-EPN-RELA and ST-EPN-YAP1, are relevant in children. Notably, these two ST-EPN molecular groups are characterized by frequent recurrent genetic alterations that are not present in any of the other groups. ST-EPN-RELA tumors mostly harbor a fusion between the NF-κB effector RELA and a less characterized neighboring gene, C11orf95, as a result of a chromothriptic event on chromosome 11q[9]. In mice, the RELA fusion protein drives tumor formation in forebrain-derived neural stem cells (NSCs) in allograft or the RCAS/tv-a system models[10]. Tumor formation is accompanied by activation of NF-κB target genes, indicating that inhibition of this signaling pathway might represent a potential targeted therapeutic approach in this molecular group[9].

ST-EPN-YAP1 tumors are characterized by recurrent fusions of the Hippo pathway regulator YAP1 to either the mastermind-like protein MAMLD1 or an uncharacterized protein, FAM118B[8]. In contrast to ST-EPN-RELA tumors, no additional alterations, including homozygous CDKN2A deletions or TP53 mutations, have been observed in ST-EPN-YAP1[8,11]. Although neither YAP1-MAMLD1 nor YAP1-FAM118B fusions have been reported in other types of cancer, it is considered likely that these constitute the oncogenic drivers of infant ST-EPNs based on high frequency of YAP1 fusions in this type of cancer. The core Hippo pathway is regulated by upstream signal transduction proteins and generally limits organ growth and tumorigenesis by retaining the transcriptional cofactor YAP1 in the cytosol[12,13]. Nuclear translocation of YAP1 may promote expansion and proliferation of undifferentiated stem cells leading to formation of epithelial or soft tissue tumors[14–17]. However, the exact oncogenic function of YAP1 and YAP1 fusion proteins in EPNs remains to be investigated.

In this study, we molecularly characterize the role of YAP1 fusion proteins in primary human EPNs. In addition, we develop an electroporation-based YAP1-MAMLD1-driven ST-EPN-YAP1 mouse model. Using this mouse model, we uncover mechanistic insights into the transforming capacity of YAP1-MAMLD1 on ventricular neural precursor cells, and we identify potential avenues for targeted therapeutic intervention.

## Results

### Nuclear localization of YAP1 fusions in human ST-EPN-YAP1s.
We first analyzed whole genome DNA methylation profiling data of 45 primary supratentorial WHO grade II or III ependymomas (ST-EPNs), all of which were predicted to be ST-

EPN-YAP1 according to the recently published brain tumor classifier[18], together with a published reference cohort of ST-EPN-RELA[8]. Unsupervised clustering by t-distributed stochastic neighbor embedding (t-SNE) consistently uncovered two stable molecular groups, previously named as ST-EPN-RELA (n = 92) and ST-EPN-YAP1 (n = 45), respectively (Fig. 1a, b). The molecular groups were closely associated with specific age groups (Supplementary Fig. 1a). While both ST-EPN-RELA and ST-EPN-YAP1 were predominantly found in pediatric patients, ST-EPN-YAP1 tumors were mostly restricted to infancy (median age of 1 year compared to 8 years in ST-EPN-RELA). Given the known role of YAP1 as a transcriptional co-activator in the nucleus as an oncogene[13,19,20], we first examined expression of YAP1 in human primary ST-EPN-YAP1 samples (Fig. 1c). Immunostaining with an anti-YAP1 antibody revealed predominant nuclear expression of the YAP1 protein in tumor cells (Fig. 1d, e). Direct phosphorylation of YAP1 on serine residue 127 (S127) by LATS1/2 normally retains the protein in the cytosol due to sequestration by the 14-3-3 protein[13]. Immunohistochemistry (IHC) with a p-YAP1 (S127)-directed antibody, however, clearly showed a nuclear localization signal (NLS) of p-YAP1 (S127) in a large fraction of cells in ST-EPN-YAP1 (Fig. 1f). These findings suggest that the phosphorylation-dependent subcellular localization machinery is dysregulated in ST-EPN-YAP1.

RNA-seq-based transcriptome analyses from seven independent ST-EPN-YAP1 human tumors predicted that 32 ± 15% of YAP1 transcripts originated from YAP1-fusion gene(s) (the mean ± S.D., n = 7, Supplementary Fig. 1b). The overall YAP1 expression level in human ST-EPN-YAP1 tumors was not higher than in other intracranial molecular ependymoma groups (Supplementary Fig. 1c), but was higher than average in comparison with other genes within the individual tumors (Supplementary Fig. 1d). Consistent with these observations, western blotting (WB) analyses revealed a comparable level of endogenous YAP1 wild-type protein across human primary ST-EPNs (Fig. 1g, h). We detected both fusion types, YAP1-MAMLD1 (140 kDa) and YAP1-FAM11B (120 kDa) in ST-EPN-YAP1 samples but not in ST-EPN-RELA samples (Fig. 1g–i). Protein levels of YAP1-MAMLD1 (Fig. 1g) and YAP1-FAM118B (Fig. 1h) were several folds higher in the nuclear fraction compared to the cytoplasmic fraction (Fig. 1i). In addition, IHC-based detection of nuclear localization of p-YAP1 (Fig. 1f) was also validated by WB (Fig. 1j). These results strongly imply a functional role of the YAP1 fusion proteins in the nucleus.

### YAP1 fusion needs MAMLD1 domain for nuclear translocation.
It has been shown that YAP1 nuclear translocation is required to exert its oncogenic function[21]. Therefore, we next investigated the mechanism underlying S127 phosphorylation-independent nuclear translocation of YAP1 fusion proteins in the developing mouse brain. In order to examine subcellular localization in vivo, we used an in utero electroporation-based gene transfer approach[22,23] (Fig. 2a). We designed experimental constructs encoding the most frequent fusion type, YAP1-MAMLD1, as well as control constructs including wild-type genes of the fusion partners and truncated and mutated YAP1. Each construct was fused to the hemagglutinin (HA)-tag under a CAG promoter and upstream of IRES-EGFP in the pT2K-based expression vector[22] (Fig. 2a). Since ST-EPN-YAP1 tumors arise in the ST region of the brain and are thought to originate from radial glia NSCs or ependymal precursor cells[24], we targeted the cerebral ventricular zone during embryogenesis. Subcellular localization of HA-tagged recombinant proteins was analyzed two days after electroporation into the lateral ventricle of E13.5 mice (Fig. 2b, c). While exogenous wild-type YAP1 protein (YAP1-HA) was predominantly

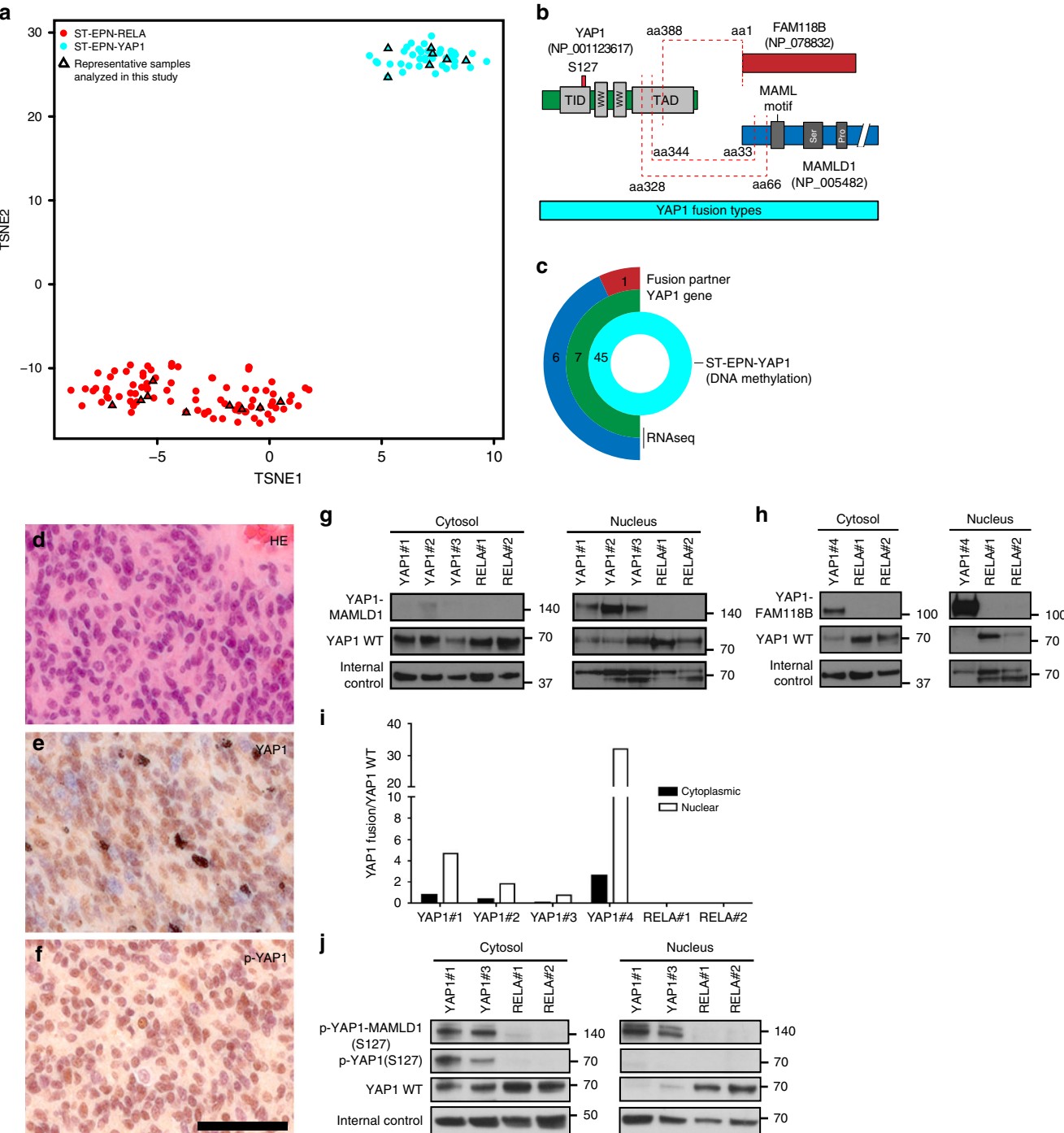

**Fig. 1** YAP1 fusion proteins are predominantly localized in the nucleus of ST-EPN-YAP1s. **a** Unsupervised 2D representation of sample correlations based on DNA methylation data by t-distributed stochastic neighbor embedding (t-SNE) dimensionality reduction. Individual samples ($n = 137$) are color-coded in the respective class color for ST-EPN-RELA (red) and ST-EPN-YAP1 (cyan). Representative samples further analyzed in this study are indicated by triangles. **b** YAP1 fusion types at protein level. Red dashed lines indicate fusion sites. Proteins are drawn to scale. TID TEA domain-containing factor-interaction domain for TEAD binding, WW protein–protein interaction domain, TAD transcriptional activation domain for TEAD, MAML mastermind-like domain, Ser serine-rich region, Pro proline-rich region. **c** Graphical summary of the ST-EPN-YAP1 cohort ($n = 45$) analyzed in this study, classified according to Genome-wide DNA methylation profiles. RNA sequencing data were available for seven samples. Absolute numbers of fusion partners are indicated. Color codes are identical to **b**. **d**–**f** Representative micrographs show **d** haematoxylin and eosin staining (H&E), immunostaining for **e** YAP1 and **f** phosphorylated YAP1 (S127) (p-YAP1) for human ST-EPN-YAP1 tumors (scale bar, 50 μm). **g, h** Detection of **g** YAP1-MAMLD1 (140 kDa) or **h** YAP1-FAM118B (120 kDa) fusions with YAP1 wild-type (WT) (75 kDa) in ST-EPN-YAP1 (YAP1#1-4) and ST-EPN-RELA tumors (RELA#1-2) by western blotting of cytosolic and nuclear fractions with an antibody recognizing the C-terminus of YAP1 protein. Actin and LaminB1 were used as an internal control for cytoplasmic and nuclear fractions. **i** Quantification of protein levels in human primary ST-EPNs. The values were normalized to the respective internal controls. **j** Detection of phosphorylated YAP1-MAMLD1 (140 kDa) in the nucleus of human ST-EPN-YAP1 tumors. Of note, phosphorylated YAP1 protein was not detected in the cytoplasm nor the nucleus in human ST-EPN-RELA tumors

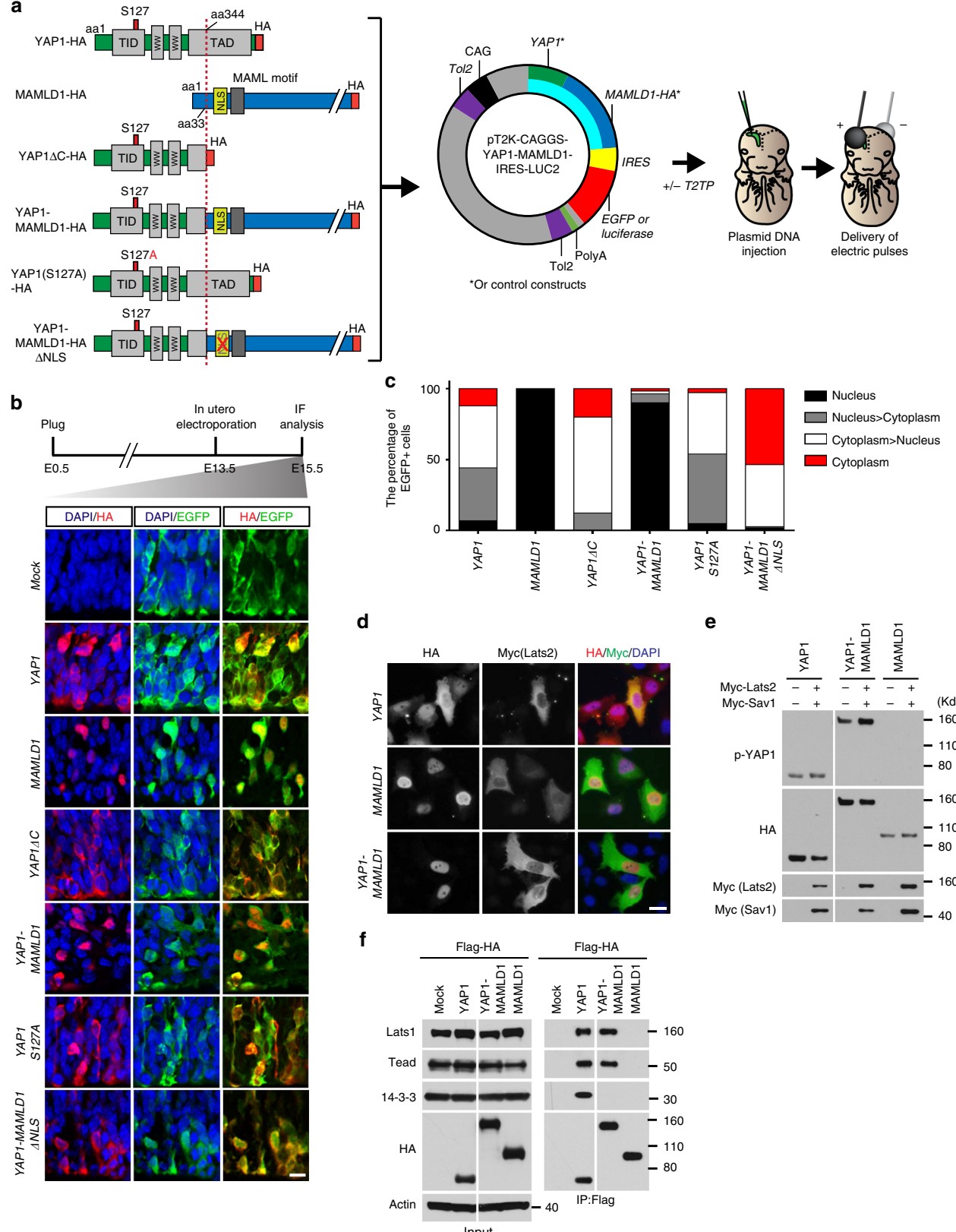

localized in the cytoplasm, exogenous wild-type MAMLD1 protein (MAMLD1-HA) accumulated in the nucleus, consistent with a previous report[25]. The exogenous full-length YAP1-MAMLD1 fusion localized within the nucleus, similar to human ST-EPN-YAP1 tumor cells. Truncated YAP1 protein (YAP1ΔC-HA),

corresponding to the complete YAP1-encoded part of the fusion protein was not detected in the nucleus. This result suggests that loss of the YAP1 C-terminus caused by formation of the fusion is not responsible for the nuclear translocation of the YAP1-MAMLD1 fusion protein.

**Fig. 2** MAMLD1 confers the nuclear translocation ability of YAP1-MAMLD1 fusion protein. **a** YAP1 fusion and control constructs at protein level and graphical illustration of the workflow used in this study. Red dashed line indicates fusion site. All constructs are tagged with the human influenza hemagglutinin surface glycoprotein (HA). Color codes are identical to Fig. 1b. The YAP1-MAMLD1 fusion or indicated control constructs were cloned into the pT2K transposable vector and injected with or without the Tol2 transposase into the lateral ventricle of E13.5 wild-type mice followed by transfection using an electroporation based in vivo gene transfer approach. NLS nuclear localization signal. CAG CMV early enhancer/chicken β actin promotor, IRES internal ribosomal entry site, Tol2 Tol2 transposase cis element. **b** Immunofluorescence micrographs for subcellular localization of indicated proteins in the cells of the ventricular zone 2 days after in utero electroporation. Mock represents the pT2K IRES-EGFP empty plasmid. Double staining was performed for DAPI/HA (left panel) or EGFP/DAPI (mid-panel) or HA/EGFP (right panel) (scale bar, 10 μm). **c** Quantification of the percentage of the electroporated cells expressing indicated exogenous (HA-tagged) proteins in the nucleus and the cytoplasm. **d** Immunostaining of LN229 cells co-expressing Myc-tagged Lats2 and indicated genes tagged by HA. The cells were stained with anti-HA and anti-Myc antibodies followed by counterstaining with DAPI. **e** Western blot of protein lysates of LN229 cells transfected with indicated genes. **f** Immunoprecipitation of protein lysates of the transfected LN229 cells with the Flag antibody followed by western blotting with indicated antibodies. Actin is used as an internal control

To explore the contribution of the MAMLD1 domain to the nuclear localization of YAP1-MAMLD1 fusion protein, we identified 11 amino acids (aa79-89, YPNKIKRPCLE) as the putative NLS of MAMLD1 using an NLS mapper tool[26]. Indeed, YAP1ΔC fused to this NLS (YAP1ΔC-MAMLD1(NLS)) restored nuclear accumulation of the protein in the 3T3 mouse fibroblast cell line (Supplementary Fig. 2). Meanwhile, YAP1-MAMLD1ΔNLS, which lacks the NLS of MAMLD1 (Fig. 2a), was predominantly distributed in the cytoplasm in vivo (Fig. 2b, c), indicating that the YAP1 fusion partner MAMLD1 drives nuclear translocation of the fusion protein via its 11 amino acid NLS.

We next investigated if the MAMLD1 domain represents a more potent nuclear shuttling mechanism for YAP1 than prevention of S127 phosphorylation. For this purpose, we introduced the S127A point mutation in the YAP1 protein (YAP1 (S127A)-HA) and examined its subcellular localization. The mutant protein accumulated in the nucleus but to a much lesser extent compared with the YAP1-MAMLD1 fusion protein (Fig. 2b, c). Of note, YAP1-MAMLD1 nuclear localization was not affected in vitro even when S127 over-phosphorylation was induced by overexpression of Lats2 and Sav1 (Fig. 2d, e), likely because the fusion was unable to interact with the cytoplasmic YAP1 regulator 14-3-3 (Fig. 2f). These data strongly suggest a phosphorylation-independent and MAMLD1-mediated nuclear localization of YAP1-MAMLD1.

To investigate the influence of the YAP1 fusion protein in differentiation of electroporated cerebral neural stem cells, we utilized a transposon-based genome integration system. Given that pT2K plasmids carry Tol2 cis-elements[27], co-transfection of the pT2K plasmids together with the Tol2 transposase (T2TP) enables integration of Tol2 cis-flanked elements, i.e. YAP1-MAMLD1 and control constructs (cf. Fig. 2a), into the genome of transfected cells and results in CAG promoter-driven constitutive expression[27]. With this electroporation strategy, we targeted NSCs that give rise to pyramidal neurons of the cortical plate and to ependymal precursor cells[28,29]. Following electroporation-based transfection with the fusion and control EGFP plasmids, the fate of transformed cells was analyzed at postnatal day (P) 7 (Fig. 3). In control animals, cells transfected with IRES-EGFP and T2TP were evenly distributed in the cortical layers and in the ventricular zone, where ependymal cells are normally located during development (Fig. 3a, b). In contrast, constitutive expression of YAP1-MAMLD1-IRES-EGFP with T2TP resulted in accumulation of EGFP-positive cells close to the cerebral ventricular zone. These cells lacked neurite outgrowth and failed to form cortical layer structures (Supplementary Fig. 3a), suggesting failure to properly differentiate and migrate (Fig. 3c). YAP1-MAMLD1-expressing cells had strong proliferation activity, as assessed by Ki67 staining, when compared to nearby non-transfected cells (Fig. 3d). This result points to a potential cell autonomous hyper-proliferation in precursor cells carrying the

fusion gene. As observed in human primary ST-EPN-YAP1 tumors (Fig. 1f, g), the phosphorylated YAP1 protein (p-YAP1), as well as pan-YAP1 protein (YAP1) was detected in the nucleus of YAP1-MAMLD1 electroporated cells (Fig. 3e–h), thus showing that its nuclear localization was independent of the S127 phosphorylation status of the N-terminal fusion partner YAP1.

**YAP1-MAMLD1 fusion drives tumor formation in vivo.** To monitor the long-term fate of transfected cells in vivo, we performed in utero electroporation of the recombinant YAP1 fusion upstream of IRES-Luciferase (Luc) or control constructs together with the T2TP transposase, allowing for in vivo bioluminescence imaging of electroporated cells (cf. Fig. 2a). Since the transforming capacity of C11orf95-RELA, a characteristic fusion of ST-EPN-RELA tumors, was reported in Cdkn2a-null embryonic NSCs[9], we tested the suitability of our electroporation system for ST-EPN tumor modeling using similar conditions. In accordance with the previous study[9], co-transfection of plasmids encoding C11orf95-RELA-IRES-Luc, and an sgRNA targeting the Cdkn2a locus (Supplementary Fig. 3b, c) with Cas9 and T2TP led to tumor formation. The electroporated mice developed neurological signs ~50 days after birth with 100% penetrance (Fig. 3i–k). This result confirmed the utility of electroporation-based gene transfer for ST-EPN modeling. Similar to C11orf95-RELA-driven animals, luciferase signal intensity in mice electroporated with YAP1-MAMLD1 steadily increased over time (Fig. 3i, j), and all animals (100% penetrance) were euthanized due to occurrence of neurological signs on average 4–5 weeks after birth (Fig. 3k). Overexpression of YAP1 wild-type (YAP1-HA), unphosphorylated YAP1 (YAP1(S127A)-HA) or truncated YAP1 (YAP1ΔC-HA and YAP1-MAMLD1ΔNLS) did not lead to tumor formation. Although we detected comparable luciferase signal intensity early after birth when compared to YAP1-MAMLD1, the luciferase signal continuously decayed and disappeared within 3–4 weeks without the occurrence of neurological signs (Fig. 3i, j). Early increased signal intensity might also relate to an initial YAP1-dependent acceleration of cellular proliferation in the cerebral ventricular zone[30]; the luciferase signal in MAMLD1 wild-type (MAMLD1-HA)-transfected animals was lost significantly faster (Fig. 3j). The control animals did not develop tumors during long-term surveillance over several months (Fig. 3k). Electroporation of YAP1-MAMLD1 in the pT2K IRES-EGFP vector revealed that tumors spread extensively through all cerebral regions (Fig. 3l, m). While most EGFP-positive cells showed proliferative activity when assessed at P7 (Fig. 2e, f), only about one-fifth (16.8 ± 2.63%) of EGFP-positive tumor cells (expressing exogenous YAP1-MAMLD1) were also proliferating (Ki67-positive) (the mean ± S.D., $n = 6$, Fig. 3n–v), potentially indicating intra-tumoral heterogeneity in established YAP1-MAMLD1-driven tumors.

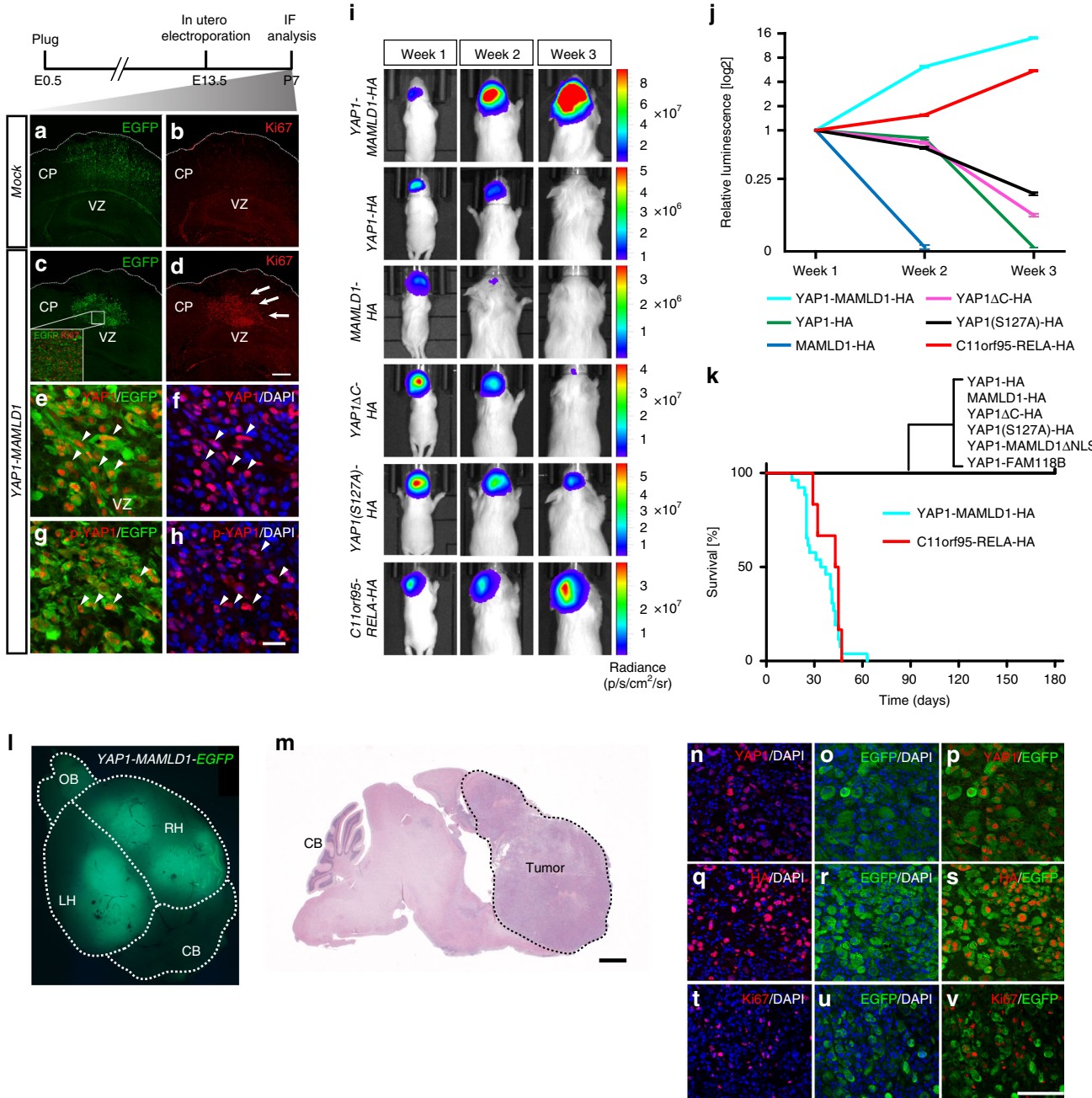

**Fig. 3** YAP1-MAMLD1 fusion drives tumor formation in vivo. **a–h** Immunofluorescence micrographs of supratentorial brain regions stained for EGFP **a**, **c** or Ki67 **b**, **d** from 7-day-old mice subjected to electroporation with YAP1-MAMLD1 or mock (EGFP alone) constructs. Insets in **c** represents electroporated cells positive for Ki67. Representative images of double staining for **e** YAP1/EGFP or **f** YAP1/DAPI and **g** p-YAP1/EGFP or **h** p-YAP1/DAPI in cells expressing the *YAP1-MAMLD1* fusion gene. Arrows in **d** indicate abnormally proliferating cells. Arrowheads in **e**, **f** and in **g**, **h** indicate the same area, respectively (scale bar in **d**, 400 μm for **a–d** and 100 μm for inset; the bars in **h**, 25 μm for **e–h**). CP cortical plate, VZ ventricular zone. **i**, **j** Luciferase-based (**i**) in vivo bioluminescence images and **j** relative changes of luminescence at weeks 1–3 after birth of animals electroporated with indicated constructs. Error bars in (**j**) indicate mean ± S.D. **k** Kaplan–Meier curves for animals electroporated with the *YAP1-MAMLD1-HA* (cyan, $n = 26$, median survival = 35.5 days), *C11orf95-RELA-HA* (red, $n = 6$, median survival = 45 days) or indicated controls ($n = 6$ for YAP1-HA, $n = 5$ for MAMLD1-HA, $n = 6$ for YAP1ΔC-HA, $n = 8$ for YAP1 (S127A)-HA, $n = 6$ for YAP1-MAMLDΔNLS-HA, $n = 8$ for YAP1-FAM118B-HA). **l** Immunofluorescence micrograph of a resected 21-day-old mouse brain electroporated with *YAP1-MAMLD1 IRES-EGFP* with the *T2TP* transposase. EGFP-positive cells have broadly spread into both the left (LH) and right (RH) hemispheres (OB olfactory bulb, CB cerebellum). **m** H&E staining of YAP1-MAMLD1-driven tumor indicated by a dotted line. CB cerebellum. Scale bar, 1 mm. **n–v** Double staining of cells derived from *YAP1-MAMLD1*-induced tumors with indicated antibodies and DAPI (scale bar, 100 μm)

**YAP1-MAMLD1-driven mouse tumors mimic human ST-EPN-YAP1.** Next, we investigated the molecular characteristics of murine tumors derived from *YAP1-MAMLD1*-electroporated animals at several developmental time points, and compared them to human counterparts. Since NSCs in the developing

cerebral ventricular zone sequentially express different transcriptional factors (TFs) as they differentiate[31] (Fig. 4a), we analyzed expression of these TFs in human EPN transcriptome data. Among human EPN subgroups, ST-EPN-YAP1 showed the highest expression of *PAX6*, a radial glial neural stem cell marker

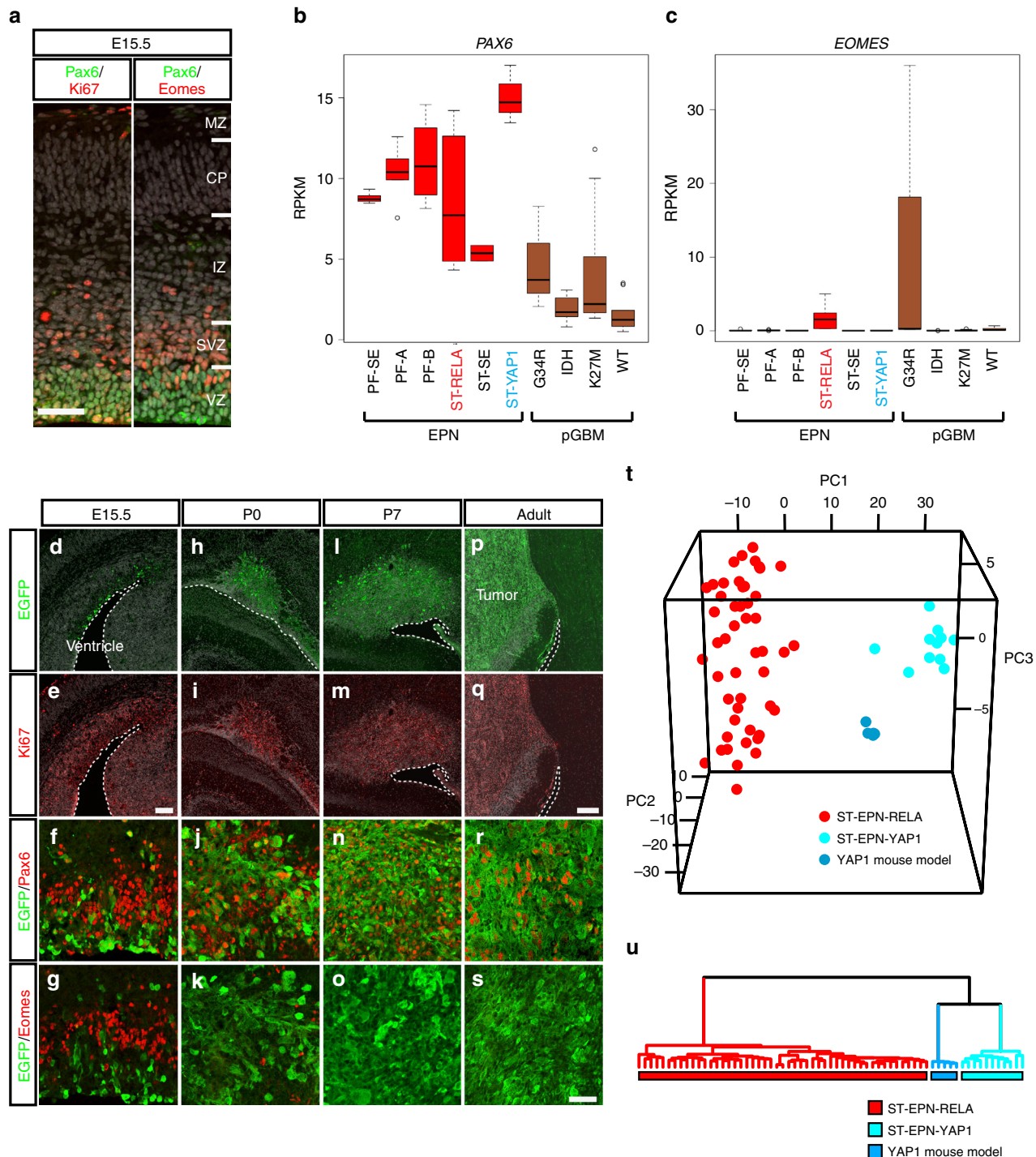

**Fig. 4** YAP1-MAMLD1-driven mouse tumors mimic molecular characteristics of ST-EPN-YAP1s. **a** IHC of E15.5 cortical plate of wildtype animals with Pax6/Ki67 (left) and Pax6/Eomes (right). MZ marginal zone, CP cortical plate, IZ intermediate zone, SVZ subventricular zone, VZ ventricular zone. Scale bar, 50 µm. **b** and **c** Box plots of transcriptome levels of **b** PAX6 and **c** EOMES across human EPN subgroups and pediatric glioblastoma multiforme (pGBMs). The center line, box limits, whiskers, and points indicate the median, upper/lower quartiles, 1.5× interquartile range and outliers, respectively. **d–s** IHC of brains electroporated with *YAP1-MAMLD1-IRES-GFP* at E13.5 at **d–g** E15.5, **h–k** P0, **l–o** P7 and **p–s** adult stages with anti-Ki67, anti-Pax6, and anti-Eomes antibodies. The margin of the ventricular zone shown in dotted lines. Scale bar in (**e**), 100 µm (for **d**, **e**), the bar in (**q**), 200 µm (for **h**, **i**, **l**, **m**, **p**, and **q**) and the bar in **s**, 50 µm (for **f**, **g**, **j**, **k**, **n**, **o**, **r**, and **s**). **t**, **u** (**t**) Principal component analysis and (**u**) hierarchical clustering based on orthologous gene expression data from human ST-EPN-RELA (red) and ST-EPN-YAP1 (cyan) tumors and YAP1-MAMLD1 fusion-driven mouse tumors (blue). Each dot represents one tumor

of the cerebral cortex[32], but did not express *EOMES*, a marker for neurogenic progenitors[31] nor *OLIG2* and *CSPG4/NG2*, markers for glial progenitors (Fig. 4b, c and Supplementary Fig. 4a, b), suggesting that ST-EPN-YAP1 could be transformed from *PAX6*-positive cells. Similar to human tumors, electroporated YAP1-MAMLD1 cells expressed Pax6 but not Eomes, Olig2, and Ng2 throughout tumorigenesis (e.g. E15.5, P0, and P7) (Fig. 4d–s and Supplementary Fig. 4c). Thus, murine tumors induced by YAP1-MAMLD1 overexpression likely arise from Pax6-positive neural stem cells. Histologically, these mouse tumors showed partly

pleomorphic areas and some necrosis without pronounced vascular changes, while growth patterns were displacing rather than infiltrative, with delineation from surrounding tissue. No perivascular pseudo-rosettes were observed (Supplementary Fig. 4d).

To test the fidelity of this new ST-EPN-YAP1 mouse model, we selected 513 genes out of the 1000 most significant differentially expressed genes arrayed on Affymetrix gene chips between human ST-EPN-YAP1 and ST-EPN-RELA samples, which were orthologous between human and mouse. Subsequently, we examined the expression levels of the genes within this orthologous set in YAP1-MAMLD1-driven mouse tumors ($n = 5$), human ST-EPN-RELA ($n = 49$) and ST-EPN-YAP1 ($n = 11$) tumors; we applied a combined unsupervised clustering and principal component analysis of all samples (Fig. 4t, u). The molecular signature of mouse tumors clearly associated with human ST-EPN-YAP1, and not human ST-EPN-RELA tumors, demonstrating molecular similarities of the mouse model to the respective human tumor samples. We further included a subset of RELA mouse models ($n = 5$) and pediatric glioblastoma multiforme (pGBM) tumors ($n = 9$) as an external control, and found that the molecular signature between the respective human and mouse EPN subgroups still exhibit the most similarity (Supplementary Fig. 4e).

**TF motifs enriched at YAP1-binding loci in ST-EPN-YAP1s.** Endogenous YAP1 wild-type mRNA and protein expression levels are comparable among molecular groups of human ST-EPNs (cf. Supplementary Fig. 1 and Fig. 1g, h). In addition, overexpression of wild-type *YAP1* did not result in tumors within the developing cerebral cortex (cf. Fig. 3k). This data prompted us to hypothesize that activation of specific YAP1 fusion-mediated gene networks causes ST-EPN-YAP1 tumor formation. To gain further insight into the transcriptional regulation caused by the YAP1 fusion, we assessed genome-wide YAP1 occupancy in fresh-frozen specimens of primary human ST-EPN-YAP1 (YAP1 fusion-positive, n = 3; Supplementary Data 1) and ST-EPN-RELA (YAP1 fusion-negative, $n = 3$; Supplementary Data 1) tumors using chromatin immunoprecipitation and subsequent DNA sequencing (ChIP-seq). YAP1 ChIP-seq was performed using an antibody against the N-terminus of the YAP1 protein to ensure recognition of the fusion protein. Data analysis revealed some loci of common YAP1 occupancy within ST-EPN-YAP1 and ST-EPN-RELA samples, but also significant differences between both groups (Supplementary Fig. 5a, b). To minimize a potential influence of the YAP1 wild-type protein, only ST-EPN-YAP1-specific high-confidence YAP1-binding sites, relating to 9% of all YAP1-binding sites within this group ($n = 1246/13,582$), were selected for further analyses (Supplementary Fig. 5b–d). YAP1 acts as a transcriptional co-activator, which is recruited to DNA via a second DNA-binding transcription factor (TF)[33]. Therefore, we next assessed TF-binding motifs within annotated EPN subgroup-specific YAP1 target loci, as well as within common YAP1 peaks shared by both ST-EPN-RELA and ST-EPN-YAP1 subgroups. Motif enrichment analysis revealed that the DNA-binding motifs of nuclear factor I (NFI) and the four TEA domain family members (TEAD1–4) were highly enriched within ST-EPN-YAP1-specific YAP1 target sites. These motifs were not among the top enriched genes in common YAP1 peaks shared with ST-EPN-RELAs nor in ST-EPN-RELA-specific YAP1-binding loci (Supplementary Fig. 5e, Supplementary Data 2). Expression levels of NFI family and TEAD1–4 were similar between ST-EPN-YAP1 and ST-EPN-RELA tumors (Supplementary Fig. 5f).

YAP1 transcriptional functions are associated with activation of *cis*-regulatory DNA elements, such as enhancers and super-

enhancers[20,34,35]. Therefore, we asked if high-confidence YAP1-binding sites consisted of group-specific enhancer or super-enhancer elements. We defined these elements by the presence of both H3K27 acetylation (H3K27ac) and YAP1 peaks (Fig. 5a, b). Approximately 87% ($n = 1084/1246$) and 45% ($n = 1100/2464$) of ST-EPN-YAP1 and ST-EPN-RELA peaks overlapped with EPN-specific enhancers ($n = 45,495$) (Supplementary Fig. 5c). By a correlation analysis of group-specific enhancer activity and gene expression, we recently identified enhancer-associated genes specifically active in ST-EPN-YAP1 tumors[5]. Using this data, 13.4% of all ST-EPN-YAP1-specific regulatory elements across the genome were bound by YAP1 ($n = 285/1246$), implying that YAP1 fusion-associated transcriptional dependencies in ST-EPN-YAP1 are restricted to relatively few specific loci. YAP1-bound enhancer and super-enhancer domains were substantially correlated with greater transcriptional load within ST-EPN-YAP1 specimens, such as in the M-phase-associated gene *NDEL1* loci (Fig. 5b, c, Supplementary Data 3).

To explore biological processes of the genes bound by YAP1 exclusively in human ST-EPN-YAP1 tumors, we applied a gene ontology (GO) analysis to 129 genes associated with the 257 YAP1-specific regulatory elements (Supplementary Data 4). GO analysis identified known YAP1-associated processes, such as cytoskeleton organization, cell migration, cell adhesion, and positive regulation of cell population among the top 10 significant biological processes in ST-EPN-YAP1 tumors (Fig. 5d), consistent with previous studies[20]. Of note, NFI and TEAD-binding sites remained the top enriched motifs when we restricted enrichment analyses to YAP1-bound regulatory elements (Fig. 5e–g and Supplementary Data 5), suggesting that these DNA-binding TFs to be of central relevance for ST-EPN-YAP1 tumor cells.

**TEADs are required for YAP1-MAMLD1-driven tumorigenesis.** Based on the ChIP-seq data from primary human ST-EPN-YAP1 specimens (cf. Fig. 5) and nuclear accumulation of the YAP1-MAMLD1 protein in human primary tumors (Fig. 1e–g), we hypothesized that the transcriptional programme of YAP1 fusion-driven ependymoma is dependent on direct interaction between the fusion and TEAD TFs. Indeed, we confirmed that expression of TEAD-mediated YAP1 targets, CTGF and CYR61[36], was strongly enhanced by YAP1-MAMLD1 compared to YAP1 WT in vitro (Fig. 6a). This interaction data is consistent with the comparative transcriptome profiles analysis between ST-EPN-YAP1 and ST-EPN-RELA subgroups, which showed that these genes are YAP1-specific in both our mouse model and human tumor samples (Supplementary Data 6). To investigate whether the YAP1–TEAD interaction is required for tumor formation in vivo, we generated a pT2K plasmid vector encoding a YAP1-MAMLD1 fusion gene mutated at the TEAD interaction site (YAP1(S94A)-MAMLD1). We confirmed that the S94A mutation effectively prevented binding of TEAD proteins to the fusion protein in vitro (Fig. 6b). Consistent with these observations, the YAP1(S94A)-MAMLD1-mutated protein failed to upregulate YAP1-TEAD target proteins (Fig. 6c).

We introduced the *YAP1(S94A)-MAMLD1* gene together with the *T2TP* transposase via in utero electroporation of mice at E13.5. Positive luciferase signals early after birth indicated successful electroporation, but the signals were lost in all animals ($n = 8$) within 3 weeks (Fig. 6d). During long-term surveillance, no tumor was detected, and none of the animals developed neurological signs (Fig. 6e), indicating that interaction between YAP1 and TEAD proteins is required for tumorigenesis. These results strongly support our insights inferred from YAP1 ChIP-seq data of human ST-EPN-YAP1 tumor samples. The

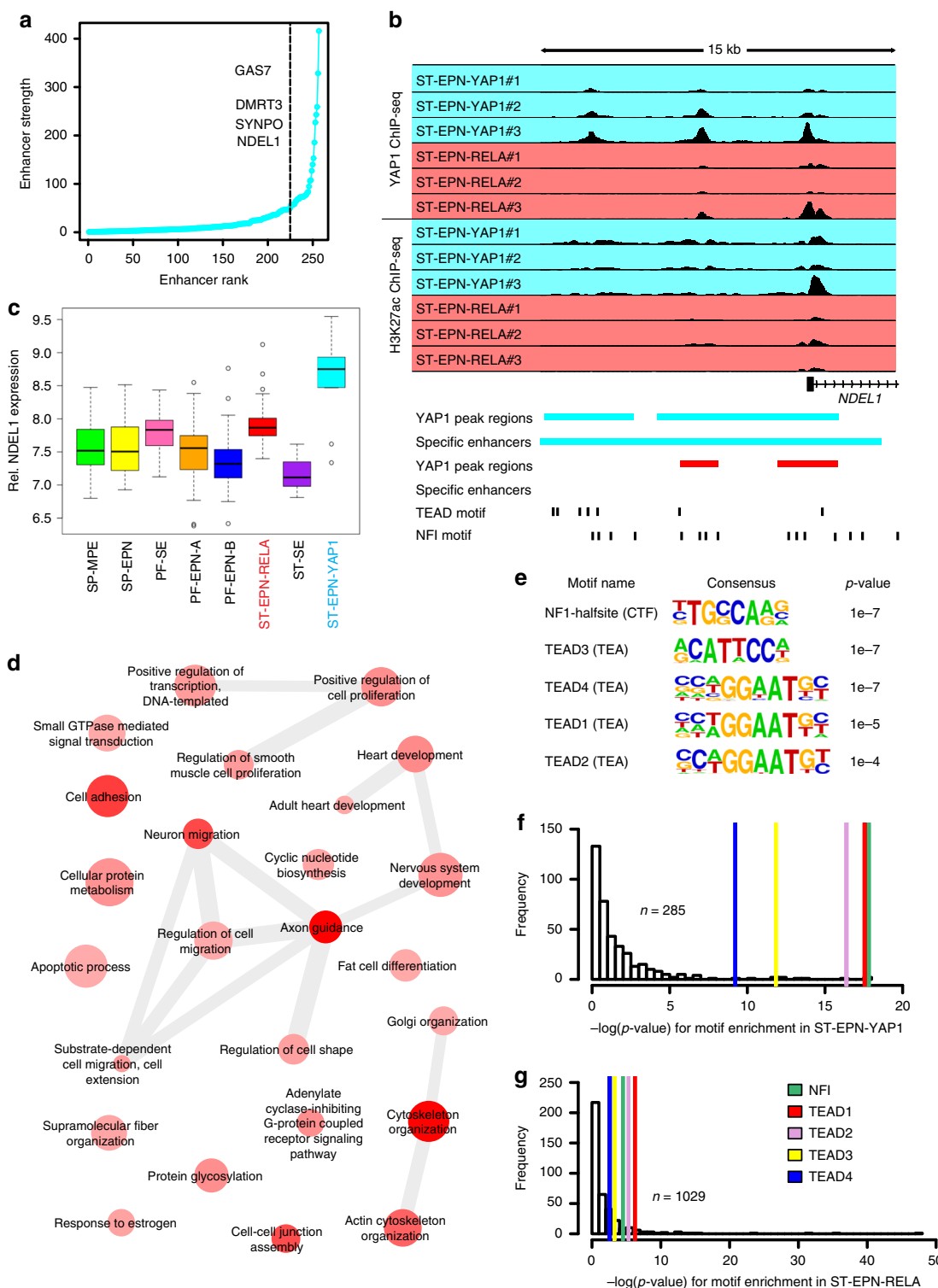

interaction between TEAD TFs and YAP1 fusion protein may act as a central mechanism for deregulation of transcriptional control and subsequent tumorigenesis and may define a specific molecular group of ST-EPNs.

**NFIs are essential for fusion-driven hyperproliferation.** While the YAP1–TEAD interaction is essential for *YAP1-MAMLD1*-driven tumorigenesis (Fig. 6), electroporated animals carrying a putative NLS-conjugated YAP1 domain (YAP1ΔC-MYC(NLS)) as

well as YAP1-FAM118B did not develop tumors despite their nuclear localization (Fig. 3k; Supplementary Figs. 3d, e and 6), implying that the MAMLD1 domain may have other roles in transforming primary cells, in addition to nuclear shuttling. Due to the fact that the NFI TF-binding motif was identified as the most enriched motif within YAP1-binding sites in human ST-EPN-YAP1s (Fig. 5), we hypothesized that NFI TFs may act as cofactors for YAP1-MAMLD1 recruitment to genomic loci and required for its oncogenic activity. Indeed, co-immunoprecipitation experiments revealed a physical interaction of both NFIA and NFIB with

**Fig. 5** TEAD and NFI motifs are enriched in YAP1-bound *cis*-regulatory elements in ST-EPN-YAP1s. **a** Ranked group-specific enhancers that overlap with group-specific YAP1 peaks. Selected genes with highest correlation in respective topology-associated domains of enhancers are indicated. Right side from dashed line represents super-enhancers. **b** YAP1 and H3K27ac ChIP-Seq data across human primary ST-EPN-YAP1 and ST-EPN-RELA tumor samples ($n = 3$ for each entity) in the *NDEL1* locus. YAP1-binding regions and H3K27Ac-marked enhancer regions of ST-EPN-YAP1 and ST-EPN-RELA colored cyan and red, respectively. TEAD and NFI motifs are shown by black bars. **c** Box plot of *NDEL1* gene expression level across molecular groups of human ependymal tumors. The center line, box limits, whiskers, and points indicate the median, upper/lower quartiles, 1.5× interquartile range and outliers respectively. **d** Gene ontology (GO) analysis of 129 genes that showed highest correlation within topology-associated domains in ST-EPN-YAP1-specific enhancers that overlap with ST-EPN-YAP1-specific YAP1 peaks. Darker color intensity reflects a smaller *p*-value of GO term enrichment, while the bubble size reflects the term frequency. **e** Top five transcriptional factor-binding motifs enriched within YAP1 peaks specific to ST-EPN-YAP1 tumors. The enrichment *p*-values are computed by HOMER tool using binomial test. **f**, **g** Transcription factor motif enrichment in group-specific YAP1 peaks overlapping with group-specific enhancers/super-enhancers in human ST-EPN-YAP1 **f** or ST-EPN-RELA **g** tumors. Colored lines indicate location of TEAD1–4-binding motifs

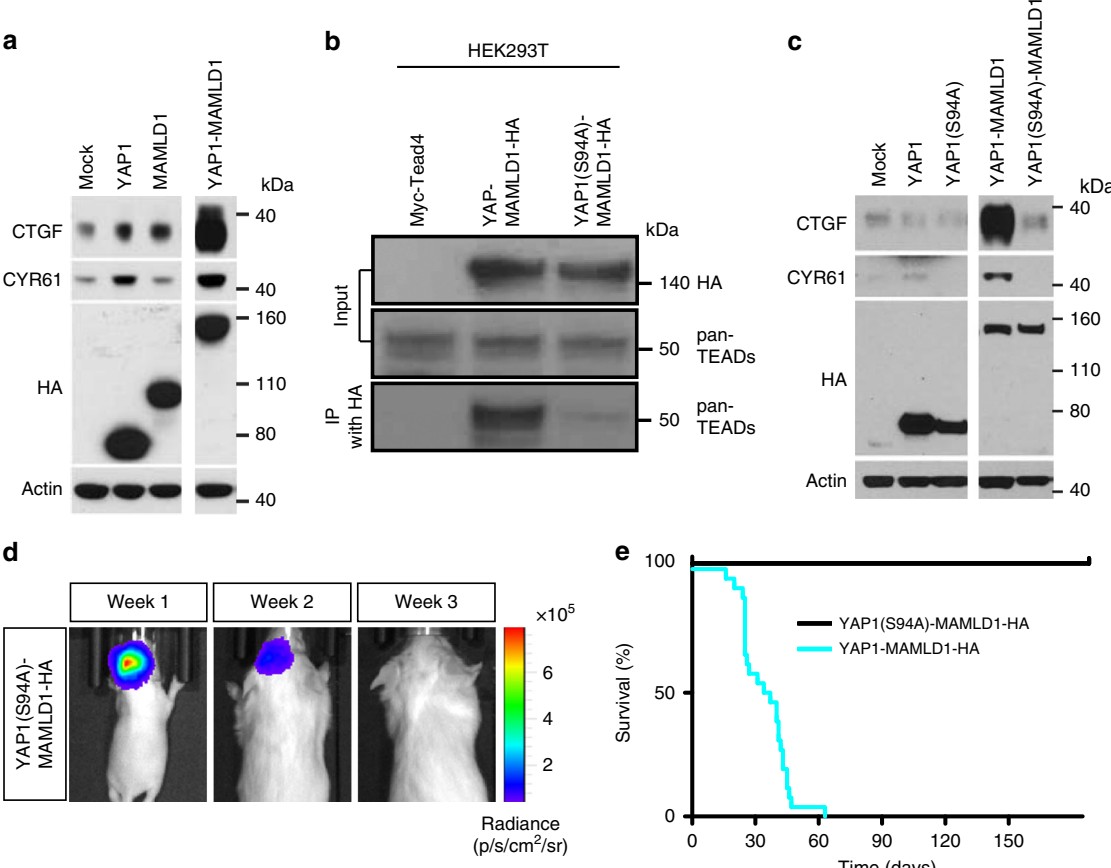

**Fig. 6** The YAP1-MAMLD1 exerts its oncogenicity via TEAD-mediated transcriptional control. **a** Western blot of human CTGF and CYR61 in LN229 cells transiently transfected with indicated genes. The transgene protein products were tagged by HA. **b** Western blot of pan-TEAD proteins (pan-TEADs) after immunoprecipitation with an anti-HA antibody for HA-tagged YAP1-MAMLD1 fusion protein, confirming that the S94A-mutated YAP1-MAMLD1 fusion has lost the ability to bind TEAD transcription factors. **c** Western blot of YAP1-Tead targets CTGF and CYR61 using NIH/3T3 cells transfected with indicated genes. The transgene protein products were tagged by HA. **d**, **e** Luciferase-based **d** in vivo bioluminescence images at weeks 1–3 after birth and **e** Kaplan–Meier curves of animals electroporated with S94A-mutated YAP1-MAMLD1 fusion. Cyan curve is same as in Fig. 3k

YAP1-MAMLD1 but not YAP1 in HEK293T cells and human primary ST-EPN-YAP1 cells (Fig. 7a, b and Supplementary Fig. 7a). ChIP-seq on human ST-EPN-YAP1s with an anti-NFIA antibody found that NFIA peaks overlapped with 96% of YAP1-interacting loci (1201 out of 1246 peaks), while this overlap was 13% in ST-EPN-RELAs (337 out of 2464 peaks) (Fig. 7c). As reported in the previous study with murine neural stem cells[37], NFIA binding was detected in the loci of TEAD-mediated YAP1 target genes *CTGF* and *CYR61* in ST-EPN-YAP1s (Supplementary Fig. 7b, c). Thus, YAP1-MAMLD1 and NFI proteins seem to interact via the MAMLD1 domain in ST-EPN-YAP1s.

Since the YAP1-bound genes with NFI-binding motifs were marked by the activating histone modification H3K27Ac (Fig.

5), we examined whether transcriptional repression of Nfi protein-target genes shared by YAP1-MAMLD1 results in reduced proliferation activity of target cells. In mice, Nfia and Nfib are expressed in the ventricular zone of the developing cerebral cortex[38,39]. Consistent with this finding, the transfected cells carrying *YAP1-MAMLD1-IRES-EGFP* expressed both Nfia and Nfib in the nucleus at 2 days post electroporation (Supplementary Fig. 7d–g). To repress Nfi target genes, we utilized Nfia-En that encodes the DNA-binding domain of Nfia fused to the transcriptional repressor domain of Engrailed and serves as an Nfi recombinant transcriptional repressor[40]. Indeed, we confirmed that Nfia-En blocked activation of *CTGF* and *CYR61* by YAP1-MAMLD1 in vitro (Supplementary Fig. 7h).

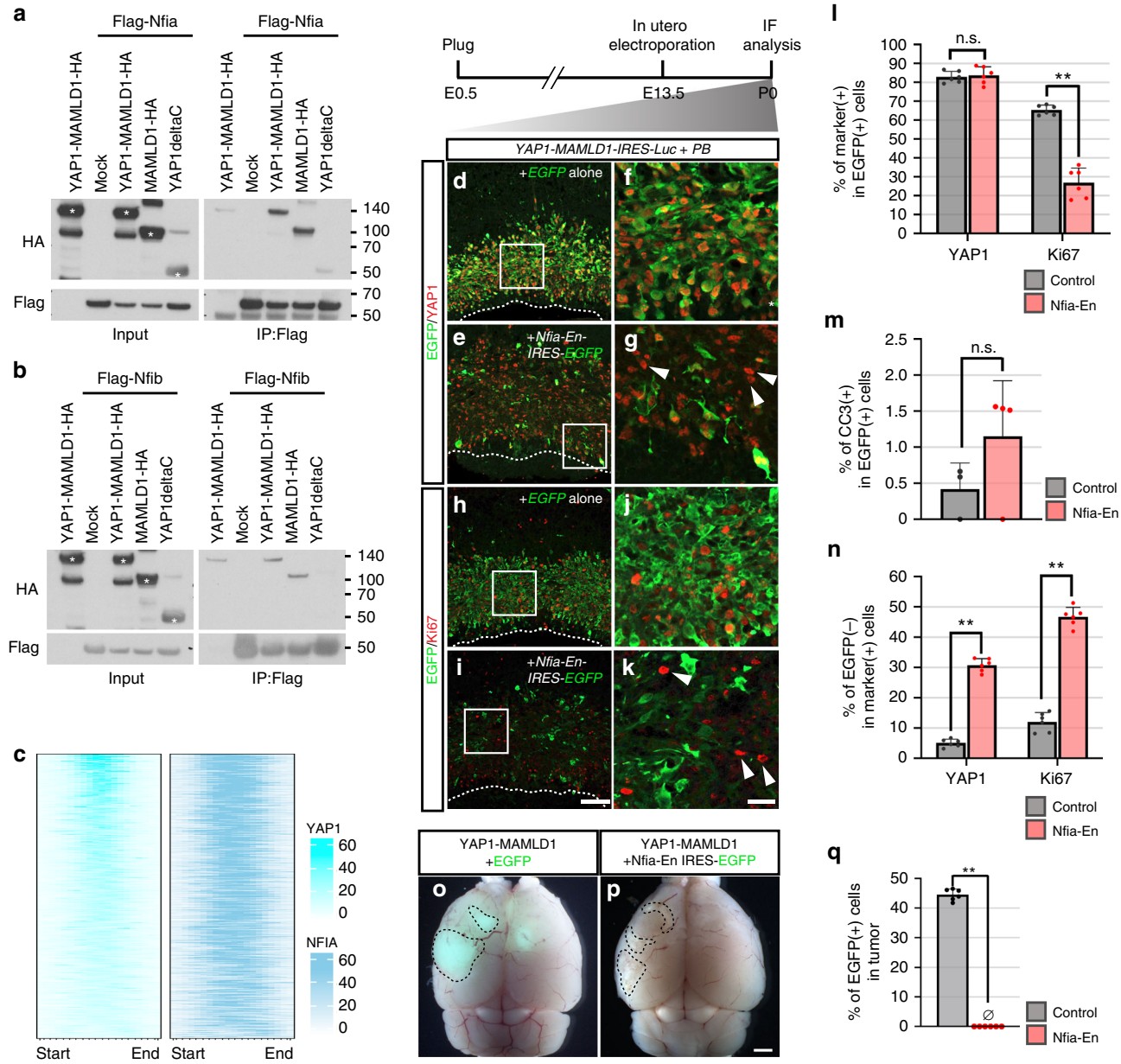

**Fig. 7** Roles of Nuclear factor I proteins in YAP1-MAMLD1-driven tumorigenesis. **a**, **b** Immunoprecipitation assay of Flag-tagged **a** Nfia and **b** Nfib and HA-tagged YAP1 fusion-related proteins in HEK293T cells. Asterisks indicate overexpressed proteins according to their predicted molecular weights. **c** An enriched heatmap of YAP1 and NFIA ChIP-seq signals within the regions of ST-EPN-YAP1 group-specific YAP1 peaks. **d–g** IHC of P0 brains with anti-GFP (green) and anti-YAP1 (red) antibodies after electroporation of *YAP1-MAMLD1-IRES-Luc* and either **d**, **f** *EGFP* alone or **e**, **g** *Nfia-En-IRES-EGFP*. Arrowheads in **g** indicate the cells expressing exogenous YAP1-MAMLD1 but Nfia-En-EGFP. **f** and **g** are high magnification views of the area outlined by a rectangle in **d** and **e**, respectively. PB Piggybac transposase. **h–k** IHC of P0 brains with anti-GFP (green) and anti-Ki67 (red) antibodies after electroporation of *YAP1-MAMLD1-IRES-Luc* and either **h**, **j** *EGFP* alone or **I**, **k** *Nfia-En-IRES-EGFP*. Arrowheads in **k** indicate abnormally proliferating cells lacking Nfia-En expression. **j** and **k** are high magnification views of the area outlined by a rectangle in **h** and **i**, respectively. Scale bar in **i** is 100 µm (for **d**, **e**, **h**, and **i**) and the scale bar in **k** is 25 µm (for **f**, **g**, **j**, and **k**). **I** The graph represents the percentages of YAP1- and Ki67-positive cells within EGFP-positive cells. Data obtained from three independent brain samples. **\*\****p* < 0.0001, n.s., not significant. The error bars indicate mean ± S.D. (*n* = 6). **m** The graph indicates the percentage of YAP1- and cleaved caspase 3-positive cells within EGFP-positive cells. *p* = 0.554. The error bars indicate mean ± S.D. (*n* = 4). **n** The graph indicates the percentage of YAP1- and Ki67-positive cells lacking EGFP expression. **\*\****p* < 0.0001. The error bars indicate mean ± S.D. (*n* = 6). **o**, **p** Dorsal views of the brains bearing tumors from the mice electroporated with *YAP1-MAMLD1-Luc* with *EGFP* (**o**) and *Nfia-Rn-IRES-EGFP* (**p**). EGFP signals are detected only in (**o**). Scale bar, 1 mm. **q** The graph shows the percentage of EGFP-positive cells in the tumor tissues. **\*\****p* < 0.0001. The error bars indicate mean ± S.D. (*n* = 6). Significant differences for **l–n**, **q** were assessed by *t*-test

Six days after in utero electroporation of *YAP1-MAMLD1-IRES-Luc* with either *EGFP* alone or with *Nfia-En-IRES-EGFP* (corresponding to P0), 82.9 ± 3.0% and 83.8 ± 4.5% of EGFP-positive cells expressed exogenous YAP1 in hyperplastic lesions of *EGFP*-only and *Nfia-En-IRES-EGFP*-expressing brains,

respectively (the mean ± S.D., Fig. 7d–g, l and Supplementary Fig. 7i–l). At this stage, significantly fewer Nfia-En-expressing cells were proliferative (Ki67-positive) than control cells (65.4 ± 2.6% for control vs. 26.5 ± 7.8% for Nfia-En, the mean ± S.D., Fig. 7h–k, l). Meanwhile, we did not observe significant

apoptotic cell death in Nfia-En-overexpressing cells at P0 (Fig. 7m and Supplementary Fig. 7m–p). In long-term observation of electroporated animals, both cases developed tumors in the cortex within a similar time period (median survival: 33 days for EGFP alone, $n = 8$ and 32.5 days for Nfia-En-EGFP, $n = 8$), probably due to the fact that not all YAP1-expressing cells in Nfia-En-electroporated mice showed expression of the EGFP-tagged inhibitory transgene (Fig. 7n). We found loss of Nfia-En-overexpressing cells in developed tumors, however ($n = 7/7$, Fig. 7o–q). Altogether, these data strongly suggest that NFI proteins are required for YAP1-MAMLD1-driven hyperproliferation of cortical neural progenitors.

## Discussion

As a clinical aspect, a more favorable outcome of ST-EPN-YAP1 compared to ST-EPN-RELA has so far only been deduced from retrospective data but still needs to be confirmed in prospective trials[8,41,42]. Although clinico-pathologic characteristics including large tumor volumes in the ventricular or periventricular region and typical appearance with multinodularity and heterogeneous contrast enhancement on MRI images in young children may hint at a ST-EPN-YAP1 tumor, a proper molecular workup is indispensable[42]. Precise molecular classification represents an important precondition for any potential future stratification that may include careful therapy de-escalation in ST-EPN-YAP1 patients and will also help to make use of subgroup-specific therapeutic targets.

Dysregulation of the Hippo/YAP1-signaling pathway has been increasingly observed as an important factor for oncogenic transformation. While amplification of wildtype *YAP1* is involved in tumor formation in various mammalian tissues[43,44], we recently identified unique YAP1 fusion variants as a characteristic hallmark of the human ST-EPN group, ST-EPN-YAP1[4,8]. The putative oncogenic function of the resulting YAP1 structural variants at an organismal level had not yet been explored. Here we show that forced expression of the most frequent ependymoma YAP1 fusion variant, YAP1-MAMLD1, is sufficient to form tumors in the developing mouse brain. *YAP1-MAMLD1*-overexpressing cortical progenitor cells fail to exit mitosis, as reported for YAP1-dysregulated satellite cells in rhabdomyosarcoma[16]. In contrast, overexpression of the YAP1 full-length cDNA did not induce tumors in our model system, suggesting a dose-independent mechanism of YAP1 fusion-driven tumorigenesis in ST-EPN-YAP1.

The nuclear-localized YAP1 protein functions as a co-transcriptional regulator with other TFs, such as TP73, RUNX, and TEAD members[33,45,46]. Disruption of its nuclear translocation machinery is often associated with cancer[21,47,48]. One such mechanism is the phosphorylation of YAP1 at Serine 127, p-YAP1(S127), by LATS1/2 kinases, which enhance cytoplasmic retention of YAP1 via 14-3-3-mediated anchoring[13]. Accordingly, deleterious somatic mutations or hyper-methylation of both *LATS1/2* and their upstream kinases *MST1/2* led to nuclear translocation of YAP1, a tumor-inducing mechanism for various cancer types[49]. Notably, we detected p-YAP1(S127) in the nucleus of human primary ST-EPN-YAP1 tumors. Consistent with this observation, we found that nuclear localization of YAP1-MAMLD1 fusion is S127 phosphorylation-independent in mice (Fig. 3g). Importantly, S127A mutation in YAP1 caused translocation into nuclei of cortical neural progenitors in vivo to much less extent than YAP1-MAMLD1 (Fig. 2b, c). Thus, phosphorylation-dependent YAP1 nuclear localization may be distinct between different cell types. In agreement with this idea, no amplification of YAP1 or mutations in S127 of YAP1 has been reported in ST-EPN-YAP1s so far.

Even though the MAMLD1 domain mediates nuclear translocation of the YAP1 domain, MAMLD1 may still be actively contributing to tumorigenesis with YAP1 in the context of ST-EPNs in other ways. This notion is supported by our findings that forced expression of YAP1(S127A), YAP1ΔC-MYC(NLS) did not cause tumors within the cerebral cortex (Fig. 3 and Supplementary Fig. 6). This data is in contrast to the oncogenic effects of YAP1(S127A) in other tissues, including liver, skin, or skeletal muscle[14–17]. While MAMLD1 functions as a co-activator of canonical Notch signaling by transactivating the *Hes3* promoter[25], we saw neither upregulation of *Hes3* nor specific binding of YAP1 to *Hes3* loci in ST-EPN-YAP1 tumors, compared with ST-EPN-RELAs. The L103P point mutation in the MAMLD1 domain of the fusion protein causes loss of its transactivation function for *Hes3*[25]. Electroporation of *YAP1-MAMLD1(L103P)* did not attenuate the YAP1-MAMLD1-mediated tumor formation ($n = 5/5$, median survival: 31.5 days), suggesting that MAMLD1 transactivation function for Notch signaling may not be relevant to the YAP1-MAMLD1 oncogenic capacity. Instead, we found that the MAMLD1 domain interacts with NFI TFs, and the fusion protein is recruited to enhancer regions enriched for TEAD and NFI-binding motifs in human ST-EPN-YAP1s.

The lack of primary tumor-derived cell lines and PDX models for ST-EPN-YAP1 tumors has hampered identification of potential drug candidates for this disease. An alternative strategy for conducting meaningful preclinical studies includes generating tumor models that exogenously introduce disease-relevant oncogenic alterations. Here we established the first ST-EPN-YAP1 mouse model that grows tumors from mid-embryonic stages with 100% penetrance, which may be utilized for future in vivo studies. In vitro studies for other types of YAP-driven cancers have proposed reduction of YAP1 dosage effects to induce tumor regression[50–52]. However, repression of YAP1 expression in human patients is currently not feasible, due to a lack of efficient targeted gene delivery methods into malignant cells. Small molecule modulators of the Hippo pathway, such as epinephrine or dobutamine, impair YAP1 function[53,54]. The mechanism of action requires phosphorylation of S127 in YAP1, leading to increased cytoplasmic retention of YAP1 and attenuation of YAP1 function in the nucleus. However, phosphorylation-independent nuclear localization of the YAP1 fusion protein in our study suggests that these molecules may be ineffective against ST-EPN-YAP1. Perhaps pharmacologic disruption of the YAP1–TEAD interaction represents a potential future strategy to efficiently treat this infant disease. Regardless, YAP1-TEAD inhibitors[55–57] that effectively penetrate the blood–brain-barrier remain to be identified. A recent study also identified BRD4 as a cofactor of the YAP1/TAZ complex and showed efficient regression of mammary and lung tumors in vivo with BET inhibitors[58], suggesting BET inhibitors may have a similar effect on ST-EPN-YAP tumors.

This study demonstrate the oncogenic function of the YAP1-MAMLD1 fusion in the development of ST-EPNs, and the data support the idea that YAP1 fusion inhibition may induce differentiation and cell-cycle exit in deregulated cerebral progenitor cells. Our integrated approach, including electroporation-based tumor modeling, provides a framework for target identification in other cancers with putative oncogenic alterations that are difficult to treat.

## Methods

**Human materials**. Tumor samples were collected after patients provided written informed consent, according to protocols approved by the institutional review boards of the University Hospital Heidelberg and the NNBurdenko Neurosurgical Institute. Only ST EPN (WHO grade II) and ST anaplastic EPN (WHO grade III) that were confidently predicted to be ST-EPN-YAP1 or ST-EPN-RELA tumors

were included in this study. This classification was made according to a DNA methylation-based CNS tumor classification approach[18]. No patient underwent chemotherapy or radiotherapy prior to the surgical removal of the primary tumor.

**Animal husbandry**. CD-1 mice were obtained from Charles River and housed in a vivarium with a 12-h light/dark cycle. The day of the plug and the birthdate are designated as embryonic day (E) 0.5 and postnatal day (P) 0, respectively. All animal experiments for this study were conducted according to the Penn State University Institutional Animal Care and Use Committee and the animal welfare regulations approved by the Animal Care and Use Committee of the National Institute of Neuroscience, NCNP, Japan and the responsible authorities in Germany (Regierungspräsidium Karlsruhe, approval number: G204/16).

**Cell lines**. HEK-293T (CRL-3216), LN229 (CRL-2611), and NIH/3T3 (CRL-1658) cells were purchased from ATCC. HEK293T cells were cultivated with Dulbecco's modified Eagle media (DMEM, Gibco) supplemented with 10% fetal bovine serum (FBS, Gibco), 100 U/mL penicillin and 100 µg/mL streptomycin. Cultures were maintained in a humidified 5% $CO_2$ atmosphere at 37 °C and subcultured when 80% of confluence was reached. Mycoplasma contamination was assessed periodically by GATC/Eurofins.

LN229 cells were cultured in DMEM (Corning, 10-013-CV) supplemented with 10% FBS (Gibco, 10437028) and 1% antibiotic–antimycotic solution (Corning, 30-004-CI) at 37 °C with 5% $CO_2$. NIH/3T3 cells were cultured in Dulbecco's modified Eagle's medium (DMEM) (Corning, 10-013-CV) supplemented with 10% bovine serum (Gibco, 16170078) and 1% antibiotic–antimycotic solution (Corning, 30-004-CI) at 37 °C with 5% $CO_2$. None of these cell lines were listed in the database of misidentified cell lines maintained by ICLAC and NCBI Biosample. These cell lines were not authenticated in this study. All cell lines were examined to be mycoplasma negative before experiments. Unless otherwise indicated, experiments were performed with cells grown to 50% confluency.

**Plasmids**. The full or partial coding regions of human *YAP1* (NM_001130145.2) and *MAMLD1* (NM_005491.4) cDNAs with a C-terminal HA tag were amplified by PCR and cloned into pT2K IRES-EGFP and pT2K IRES-Luc plasmid vectors[22]. The HA-tagged *YAP1-MAMLD1* fusion gene corresponding to the fusion gene expressed in ST-EPN-YAPs was synthesized by Life Technologies (Darmstadt, Germany) and inserted into the pT2K expression plasmids. *YAP1(S127A)* and *YAP1(S94A)-MAMLD1* were generated from *YAP1* and *YAP1-MAMLD1* by site-directed mutagenesis. For Tol2-based stable gene expression, pT2K plasmids were co-transfected with *Tol2 transposase* encoded in the pCAGGS plasmid[27]. The Myc-tagged Tead4 expression vector was obtained from Addgene (#24638). px330-based sgRNAs were used for induction of loss of function mutations in *Cdkn2a*. For SURVEYOR assay, a 500 bp DNA fragment spanning the targeting region of *Cdkn2a* was amplified by PCR. Primers used for PCR: 5′-CGGCGATGTTCTA CAGGAG-3′ and 5′-GAAGCTATGCCCGTCGGTC-3′. Piggy bac (PB)-based expression vectors were generated by cloning the CAG-IRES-EGFP cassette from pCAG IRES-EGFP[59] into the pPB EGFP[60] with NotI. Nfia-En cDNA[40] was inserted into the XhoI site of the multiple cloning site, yielding pPB Nfia-En-IRES-EGFP. For generation of YAP1-MAMLD1ΔNLS cDNA, putative NLS sequences were predicted by examining the sequence of MAMLD1 using a NLS mapper tool[26]. YAP1-MAMLD1 cDNA was subjected to site-directed mutagenesis using the Quikchange mutagenesis Kit (Agilent).

**Antibodies**. The following antibodies were used in this study: YAP1 (GTX129151, GeneTex, 1:500, for IHC), YAP1 (#14074, CST, 1:1000 for WB and 1:200 for IHC), YAP1 (sc-15407×, SCBT, 1:1000 for WB and ChIP), p-YAP(S127) (ab76252, abcam, 1:200 for IHC), p-YAP(S127) (#13008, Cell signaling, 1:1000 for WB and IHC, as validated in Supplementary Fig. 8), Ki67 (ab15580, Abcam, 1:500 for IHC), HA (#3724, CST, 1:500 for ICC and IHC), HA (#MMS-101P, BioLegend, 1:2000 for WB), Myc (#9E10, Developmental Studies Hybridoma Bank, 1:500 for WB), Lats1 (#3447, CST, 1:1000 for WB), Actin (A5316, Sigma-Aldrich, 1:5000 and ab49900, abcam, 1:25000 for WB), CTGF (sc-14939, SCBT, 1:1000 for WB), Cyr61 (sc-13100, SCBT, 1:1000 for WB), 14-3-3 (sc-629, SCBT, 1:1000 for WB), FLAG (M2, Sigma, 1:100 for IP, 1:1000 for WB), Pan-Tead (#13295, CST, 1:1000 for WB), TBR2/Eomes (ab23345, abcam, 1:500 for IHC), NG2 (ab129051, Abcam, 1:400 for IHC), PAX6 (PPRB-278P, Covance, 1:500 for IHC; Cat#sc-81649, SCBT, 1:200 for IHC), Olig-2 (AB9610, Millipore, 1:500 for IHC), Nfia (HPA008884, Sigma, 1:250 for IHC and 1:500 for WB), Nfib (HPA003956, Sigma, 1:250 for IHC and 1:500 for WB), Cleaved Caspase 3 (#9664, CST, 1:100 for IHC) and GFP (#ab13970, Abcam, 1:1000 for IHC).

**Immunostaining**. E15.5, P0 and P7 brains from electroporated mice were dissected and fixed with 4% PFA/PBS at 4 °C overnight. After cryoprotection with 30% (w/v) sucrose in PBS, the brains were embedded in OCT compound for frozen blocks. 12 µm-thick cryosections were processed by heat-induced epitope retrieval in citrate buffer (pH 6.0) before IHC. 4 µm-thick paraffin-embedded human and murine tumor sections were immunostained according to the procedures in published protocols[61]. After deparafinization, the sections were incubated with the primary antibodies at room temperature (RT) overnight, followed by

administration with biotinylated secondary antibodies (Vector) at RT for 1 h. The signals were amplified by a horseradish peroxidase system (ABC kit, Vector) followed by DAB staining (Sigma-Aldrich).

For immunofluorescence, sections were blocked with 10% normal donkey serum (NDS) in PBS 0.1% Triton-X (PBST) for 30 min at RT and incubated with the primary antibody overnight at 4 °C. After several washes with PBST, the sections were incubated with the secondary antibody diluted with 10% NDS in TBST for 1 h at RT. Slides were mounted in ProLong Gold Mountant (Invitrogen #P10144). Nuclei were stained with DAPI (300 nM). Images were acquired with confocal microscopes (ZEISS LSM 800 and Leica SP8 LIGHTNING).

For the quantification of the HA nuclear localization in E15.5 electroporated samples, a 0.1 mm² selection box was selected as the counting area. The selection box was kept constant throughout the analysis. The cells double-positive for EGFP and HA were counted and separated into four categories based on the location of the HA signal: (1) nucleus: where HA signal was located exclusively in the nucleus, (2) nucleus>cytoplasm: where the fluorescence intensity was higher in the nucleus than the cytoplasm, (3) cytoplasm>nucleus: where fluorescence intensity was higher in the cytoplasm than the nucleus, and (4) cytoplasm: where HA signal was solely detected in the cytoplasm. Approximately 100–300 cells were counted for each sample. The data is presented as percentage of cells for each category (HA localization category/total cells nuclei × 100).

For proliferation quantification in developed tumors, a 6 mm² area was selected in the tumor center. Two sections per tumor were quantified in three independent tumor samples. The sections were stained with EGFP and Ki67 and the EGFP(+) Ki67(−), EGFP(+)Ki67(+) and EGFP(−)Ki67(+) cells were counted manually using ImageJ (Maryland, USA). Only clearly labeled cells with a minimum five foci were included. Around 300–400 cells were counted in total for each tumor. The data is shown as percentage of positive cells = (positive nuclei cells/total cells nuclei × 100).

For proliferation quantification of Nfia-related constructs, two sections per P0 electroporation sample were used from three independent electroporation samples. The sections were stained with EGFP and Ki67 and analyzed. A 0.12-mm² selection area was used to define the counting area of the tissue. EGFP(+)Ki67(+) and EGFP (+) cells were counted manually using ImageJ. More than 100 cells were counted in total for each tumor section. The proliferation index is calculated as double positive cells/total EGFP-positive cells × 100. For cleaved Caspase-3 quantification, one section per P0 electroporated sample was counted from three independent electroporated samples. Sections were stained with anti-EGFP and anti-CC3 prior analysis. An area of 0.1 mm² was delimited in each section and the total of CC3 ⁺EGFP⁺ and EGFP⁺ cells located in the area were counted manually using Image J. Approximately 300 cells were counted per section. Apoptotic index was calculated using the following formula: CC3(+)EGFP(+)/EGFP(+) × 100.

For quantification of EGFP-positive cells inside *YAP1-MAMLD1*-driven tumors, one section per adult mouse electroporated at E13.5 that developed neurological signs was counted in six of arbitrarily selected fields from two independent mice. Stained sections with anti-EGFP and DAPI were submitted to imaging acquisition at a confocal microscopy and tile images were obtained. An area of 0.05 mm² was selected in the center of the tumor and total of EGFP⁺ cells were counted. All cells counterstained with DAPI within the selected area were manually counted using ImageJ. Quantification data is presented as percentage of EGFP(+) cells.

For in vitro cultured cell staining, LN229 and NIH/3T3 cells on coverslips were fixed with 4% PFA/PBS for 20 min and incubated in permeabilization buffer (PDT: 0.3% sodium deoxycholate, 0.3% Triton X-100 in PBS) for 30 min on ice. Fixed cells were then blocked with 5% BSA/PBS at 4 °C for 1 h, followed by incubation overnight at 4 °C with primary antibodies diluted in 2.5% BSA/0.05% Triton X-100/PBS. After washing with 0.1% Triton X-100/PBS, cells were incubated with secondary antibodies diluted in 2.5% BSA/0.05% Triton X-100/PBS for 2 h at 4 °C. Cells were then washed with 0.1% Triton X-100/PBS, rinsed with PBS, and mounted in ProLong Gold Mountant (Invitrogen #P10144). Nuclei were stained with DAPI (300 nM) where indicated.

**Western blotting**. Human primary ST-EPNs were sonicated in hypotonic solution. Nuclear and cytoplasmic fractions were separated by micro-centrifugation and lysed using NE-PER Nuclear and Cytoplasmic Extraction Reagents (#78833, Thermo Scientific) with Complete protease inhibitor cocktail (Cat#11697498001, Roche). For p-YAP1 detection, NE-PER Nuclear and Cytoplasmic Extraction Reagents was supplemented with Complete protease inhibitor cocktail (Cat#11697498001, Roche) and PhosSTOP™ (Cat#4906845001, Merck). To validate expression vectors in this study, HEK293T cells were transfected with the plasmids and harvested 2 days post-transfection. The cell pellets were lysed with RIPA buffer and 10 µg of the protein lysates were used for protein detection[61]. Briefly, proteins were denatured for 5 min at 95 °C, loaded on NuPAGE Bis–Tris (#NP0301BOX, Invitrogen) or NuPAGE Tris-Acetate Gels (#EA0375BOX, Invitrogen) and separated by SDS–PAGE (100–120 V). Proteins were transferred to methanol-activated PVDF membrane by tank electrotransfer in Towbin buffer for 1 h30 min at 100 V. Membrane was blocked with 5% skim-milk in 0.5% TritonX/TBS (TBST) for 1 h at RT prior overnight incubation with primary antibody. After washing with TBST, membrane was incubated with secondary antibody for 1 h at RT. For developing, the membrane was incubated with either ECL (RPN2106, GE Lifesciences) or ECL

Prime (RPN2232, GE Lifesciences) as recommended by the manufacturer followed by exposure to autoradiography films in a dark room. To study gene functions in vitro, the plasmids were transiently transfected into LN229 and NIH/3T3 cells. The cells were harvested 1 day post-transfect and lysed in SDS-lysis buffer (10 mM Tris–HCl, pH 7.5, 1% SDS, 50 mM NaF, 1 mM NaVO$_4$), followed by SDS–PAGE[62]. Uncropped immunoblotting results are shown in Supplementary Fig. 9.

**Immuno-precipitation (IP).** IP was performed according to published protocols[61]. HEK293T and LN229 cells were transfected with expression vectors for proteins of interest and harvested 2 days post-transfection. The cell pellets were lysed with either IP buffer (20 mM Tris–HCl, pH 8.0, 200 mM NaCl, 1 mM EDTA, 1 mM EGTA, 1% Triton X-100) or IP lysis buffer (Cat# 87788, Thermo Scientific) with Complete protease inhibitor cocktail (Cat#11697498001, Roche). For the Hippo pathway components IP, cells were lysed with RIPA buffer supplemented with Complete protease inhibitor cocktail. Protein extracts were incubated with anti-FLAG M2 Affinity Gel (Sigma #A2220) at 4 °C for 2 h. The immunoprecipitates were washed four times with RIPA buffer and were extracted with NuPaGE LDS sample buffer and were analyzed by WB. For NFI IP, cells were lysed with IP buffer supplemented with Complete protease inhibitor cocktail 2 days post-transfection and protein extracts were incubated with Anti-FLAG™ M2 Magnetic Beads (Cat#M8823, 30 μL/sample, Sigma-Aldrich) for 4 h at RT. Protein complexes were pulled down with a magnetic rack. After successive washing with IP buffer, the proteins trapped by the beads were extracted with NuPaGE LDS sample buffer and were analyzed by WB.

**In utero electroporation.** In utero electroporation was performed as reported previously[23,63]. Specifically, DNA plasmid mixture (1 mg/ml for each plasmid) were injected into the lateral ventricle of E13.5 embryos and square electric pulses (32 V, 50 ms-on, 450 ms-off, five pulses) were delivered using forceps-like electrodes. The operated embryos were sacrificed at E15.5, P0, P7 or when the animals exhibited neurological signs of brain tumors. EGFP-expressing brain samples were selected for further histological analysis.

For in vivo tumor formation analysis, luciferase-expressing electroporated animals were chosen at neonatal stages by intraperitoneal injection of D-Luciferin and subsequent bioluminescence imaging. Growth of transfected cells was monitored every week by measurement of intensity of bioluminescence with IVIS Lumina LT Series III Caliper (Perkin Elmer). The animals were sacrificed, once they exhibited neurological signs, such as head tilting, abnormal gait, and a hunched posture, or at 6 months of age if showing no symptoms. Total RNAs from the developed tumors were extracted with QIAGEN RNeasy Plus kit and analyzed with Affymetrix Genechip 430v2. Genomic DNA from tumors were isolated for validation of induction of somatic mutations in the *Cdkn2a* loci. PCR-amplified DNA fragments covering the target region were subcloned into the pGEM-T easy plasmid (Promega), followed by Sanger sequencing. The primer sets for PCR are 5′-CGGCGATGTTCTACAGGAG-3′ and 5′-GAAGCTATGCCCGTCGGTC-3′.

**Tumor model cross-species verification.** The comparison of mouse tumor samples to patient tumor samples was performed by applying semi-supervised clustering on Affymetrix microarray gene expression data from ependymoma tumors (YAP1: 11 samples, RELA: 49 samples). For cross-species comparison in Fig. 4t and u, differentially expressed genes between the groups were detected with Limma package[64] and sorted according to their adjusted *p*-value (<0.05). The top 1000 differentially expressed Affy probes were selected and combined based on the selection of the highest expressed probe per gene leading to 671 genes. Next, using ENCODE databases 513 human–mouse orthologous genes were selected from the 671 genes. For the selected orthologous gene, final verification was performed with unsupervised hierarchical clustering and principal component analysis of ST-EPN-RELA/ST-EPN-YAP1 human tumor data combined with five YAP1 tumor model samples.

For Supplementary Fig. 4e, differentially expressed Affy gene probes between the *YAP1-MAMLD1*-driven and *C11orf95-RELA*-driven mouse models were selected with high confidence ($n = 4581$, $p < 0.05$). The orthologues among the selected genes were compared to the differentially expressed Affy gene probes between YAP1 and RELA human tumors (top 5000 most confident probes), resulting in 539 probes representing common 308 orthologous genes (Supplementary Data 6). Additional verification of the YAP1 mouse model was performed by inclusion of Affymetrix data from human pGBM tumors ($n = 9$) and the RELA mouse models ($n = 5$) in the principal component analysis with the selected 308 genes across species.

**YAP1 and NFIA TF ChIP sequencing.** Enrichment of YAP1-bound and NFIA-bound DNA fragments by ChIP and the subsequent library preparation of the ChIP-derived DNAs by Active Motif (Carlsbad, CA)[5]. Ependymoma brain tumor tissue was submersed in PBS + 1% formaldehyde, cut into small pieces and incubated at RT for 15 min. Fixation was stopped by the addition of 0.125 M glycine (final concentration). The tissue pieces were then treated with a TissueTearer and finally spun down and washed 2× in PBS. Chromatin was isolated by the addition of lysis buffer, followed by disruption with a Dounce homogenizer. Lysates were sonicated using the EpiShear™ Probe Sonicator (Active Motif, cat # 53051) with an

EpiShear™ Cooled Sonication Platform (Active Motif, cat # 53080) and the DNA sheared to an average length of 300–500 bp. Genomic DNA (Input) was prepared by treating aliquots of chromatin with RNase, proteinase K, and heat for de-crosslinking overnight at 65 °C, followed by ethanol precipitation. Pellets were resuspended and the resulting DNA was quantified on a NanoDrop spectrophotometer. Extrapolation to the original chromatin volume allowed quantitation of the total chromatin yield. An aliquot of chromatin (15 μg) was precleared with protein A agarose beads (Invitrogen). Genomic DNA regions of interest were isolated using 4 μg of antibody against NFIA (Atlas Antibodies, catalog number HPA008884) or 8 μg of antibody against YAP1 (Santa Cruz, catalog number sc-15407×). Complexes were washed, eluted from the beads with SDS buffer, and subjected to RNase and proteinase K treatment. Crosslinks were reversed by incubation overnight at 65 °C, and ChIP DNA was purified by phenol–chloroform extraction and ethanol precipitation. Quantitative PCR (QPCR) reactions were carried out in triplicate on specific genomic regions using SYBR Green Supermix (Bio-Rad). The resulting signals were normalized for primer efficiency by carrying out QPCR for each primer pair using Input DNA.

The ChIP-sequencing reads were aligned using BWA v0.5.10 tool[65] to the GRCh37 1000G reference genome. Data quality controls were performed by employing Qualimap v2.2[66] (Supplementary Data 1). DNA interaction sites of YAP1 and NFIA were identified by applying MACS v1.4[67] with a *p*-value threshold of 1e−9 and low coverage whole genome sequencing data of matched samples as control data. Unsupervised clustering of top 5000 variable normalized YAP1 peak signals with subtracted control read counts was applied by performing principal component analysis. To identify high-confidence YAP1 DNA interaction sites common in all three tested YAP1 or RELA EPNs, respectively, we have combined YAP1 interaction sites per group using the merge function available in Bedtools v2.24[68] by requesting all three samples per group to support the presence of YAP1 DNA interaction sites. The two resulting sets of YAP1 DNA interaction sites were compared to each other to discriminate between interaction sites common and specific for YAP1 or RELA EPNs, respectively. Overlapping peak sets among the samples were visualized using UpSetR v1.3.3[69] based on the number of samples covering each peak. High-confidence NFIA peaks obtained from the subset of ST-EPN-YAPs were also merged and compared to YAP1 sites in terms of the overlaps of genomic regions.

The TF motif enrichment analysis was performed using HOMER v4.8.3[70]. Here, we have added the TEAD3 motif (ID: MA0808.1) taken from the JASPAR CORE database[71] to HOMER's default database of 388 motifs (including TEAD1, TEAD2, and TEAD4 motifs). The set of YAP1 peaks common between both groups was applied as additional control for the motif-enrichment analysis. Group-specific enhancer and super enhancer regions and the expression of their correlated target genes were obtained from the previous study[5].

**RNA-sequencing.** Sequencing reads were aligned to GRCh37 1000G reference genome using STAR 2.3.0[72] by reporting only reads with one best alignment. Gene counting of uniquely aligned reads was performed using the package Subread v1.4.6[73] based on Gencode v19 annotations. InFusion toolkit v.0.6.3 was applied for fusion gene discovery[74] with default parameter settings. The weight of YAP1 fusion in comparison to WT was calculated as a proportion of split reads covering only breakpoint between exons of genes forming the fusion compared to the total number of split reads covering the exon border including those that support WT YAP1 exon connections (see Supplementary Data 1). The comparison of gene expression in EPNs to pediatric glioblastoma was performed using RNA-seq data from the corresponding study[75].

**H3K27ac ChIP-sequencing.** The initial data analysis, peak calling, and enhancer-associated genes detection were performed in the recent study[5]. We used the same alignment and peak calling procedure for YAP1 ChIP-seq data processing. Correlation of group-specific enhancer signals with gene expression was performed based on the search inside of topologically associated domains with adjusted *p*-value limit 0.01[76].

**GO analysis.** The GO analysis of YAP1-bound genes associated with H3K27-marked enhancers in ST-EPN-YAP1 and ST-EPN-RELA tumors was performed using DAVID resources[77] with focus on Biological Process, Molecular Function, Cellular Component, and KEGG pathway categories (see Supplementary Data 4). The visualization was generated using REViGO web service[78].

**Reporting summary.** Further information on research design is available in the Nature Research Reporting Summary linked to this article.

## Data availability

The all sequencing data in this study are available in the following public databases: EGAS00001002696 for RNAseq and ChIP seq on ST-EPNs, GSE65362 for the 450K-methylation arrays for ST-EPNs, GSE64415 for the Affymetrix expression datasets of ST-EPNs, GSE134404, for the Affymetrix expression datasets of glioblastomas, and GSE110625 for the Affymetrix expression datasets of murine YAP1-MAMLD1-driven and C11orf95-RELA-driven tumors.

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

## Acknowledgements
This work was supported by the CERN Research Fellowship (to K.W.P.), the ICGC PedBrain Tumour Project funded by the German Cancer Aid (109252) and the German Federal Ministry of Education and Research (to P.L. and S.M.P.), the KIKA grant (#90) (to M.K.). the U.S. National Institutes of Health Grant K22 5K22CA190440 (to W.L.), the Four Diamonds Fund for Pediatric Cancer Research (to W.L.), and the Deutsche Forschungsgemeinschaft, KA 4472/1-1 (to D.K.). We thank Y. Takahashi (Kyoto University) and K. Kawakami (National Institute of Genetics, Japan) for Tol2 transposon related vectors; Lena Kutscher (DKFZ) for a proof-reading and editing; J. LoTurco (University of Conneticut) for pPB-eGFP vector; R. Gronostajski (State University of New York at Buffalo) for Nfia-En cDNA; D.J. Pan (UT Southwestern Medical Center) and F. Giancotti (MD Anderson Cancer Center) for vectors expressing MST2, Lats and Sav1, K. Reifenberg and K. Dell for helpful assistance for animal experiments at DKFZ; the Imaging and Cytometry, Genomics and Proteomics Core Facilities of the DKFZ and Penn State College of Medicine, and the Carl Zeiss Imaging Center in the DKFZ.

## Author contributions
K.W.P. and D.K. conceived the project. K.W.P., S.M.P., W.L., and D.K. supervised the project. K.W.P., Y.W., P.B.G.S., M.V., L.Z., L.S., M.G., M.M., T.W., M.Z., W.L., and D.K. conducted in vitro experiments. Y.W., P.B.G.S., M.V., S.B., N.M., W.L., and D.K. performed animal experiments. K.O., Y.I.-K., T.S., H.K.-Q., H.-K.L., and L.C. performed computational analysis. F.S., D.C., and A.K. performed histopathological analysis. F.A., E.H., K.M., J.B., L.J.R., D.T.W.J., P.L., M.H., S.M.P., and M.K. provided biomaterials and resources. K.W.P., K.O., M.V., W.L., and D.K. wrote an original manuscript. All the authors provided intellectual inputs and edited the manuscript.

## Additional information

**Competing interests:** The authors declare no competing interests.

