## [Peer Review File · Nature Communications]

Reviewers' comments:

Reviewer #1 (Remarks to the Author): Expert in brain cancer

An important paper describing a mouse model for YAP fusions in ependymoma. Not only do the authors generate a new model, they use this model and comparative biology with human tumors to gain insights into mechanisms of transformation by YAP fusion proteins, through activation of NFI and TEAD targets. My suggestions:

1. For nuclear localization of p-YAP1 (S127), this is an important point, and IHC staining not so clear (and sometimes not so accurate with phospho antibodies). Can the phospho antibody also be analyzed in western localization experiments, in Fig 1 h and i. Also regarding this point, S128 phosphorylation augments nuclear localization of YAP, while S127 inhibits. Can the authors check the specificity of their S127 phospho antibody by checking whether this phosphorylation is gone in the S127A mutant, in Fig 2?
2. In Fig 3, can neurite outgrowth and formation of cortical structures in control conditions be better delineated (perhaps in a supplemental figure) to better illustrate failure of these processes by fusion proteins?
3. Page 6 and elsewhere, mice develop neurological signs not neurological symptoms (needs to be corrected in 7 places), since mice can't communicate how they feel, minor issue.
4. Lettering in Fig 4 is wrong in text, PCA analysis in text refers to IF staining.
5. H and E section in Fig S5 of poor quality, can this be replaced with a figure that is more cleanly stained and photographed?

Reviewer #2 (Remarks to the Author): Yap expert

The authors demonstrate the oncogenic role of a novel YAP1-MAMLD1 fusion protein in supratentorial ependymoma. The YAP1-MAMLD1 fusion protein preferentially localizes to nucleus, which is driven by MAMLD1 independent of S127 phosphorylation. In addition, MAMLD1 facilitates the binding of YAP1-MAMLD1 fusion protein to NFI. The authors further demonstrate that YAP1, MAMLD1, and NFI are all required for the oncogenic transformation using an electron transportation model. Overall, the experiments are well designed, and properly controlled. The findings are interesting. However, some additional experiments will help strengthen the overall impacts of the manuscript.

Major comments:

1. Data in Figure 1a-c are largely re-plotting of the same data as what was published by the group in Cancer Cell 2015 except for five new samples. Figure 1c is nearly identical to Figure 2c in Cancer Cell 2015 paper. This raises potential copyright infringement issue and should be removed from the current manuscript.
2. Figure 1h: It is important to assess the relative expression levels of YAP1-MAMLD1 to YAP1 WT in ST-EPN-YAP1 tumors, and to YAP1 WT in ST-EPN-RELA tumors. Please repeat the western to show YAP-fusion and WT protein detection on the same blot. A p-YAP S127 blot need also be done with the same samples to demonstrate that phosphorylation-independent nuclear translocation in ST-EPN-YAP1 tumors. This is very important because all the in vivo experiments were done with exogenously expressed YAP1-MAMLD1. Also, contrary to the authors' claim in page 7, the overall mRNA levels of YAP (both fusion and WT) is significantly lower comparing to the rest of the subtypes as shown in Figure S1. It seems that YAP1 WT proteins are expressed at comparable levels between ST-EPN-YAP1 and ST-EPN-RELA tumors. If endogenous YAP1-MAMLD1 is very lowly

expressed, could it really be the oncogenic driver? Could knockdown of the YAP1-MAMLD1 fusion protein be done with primary ST-EPN-YAP1 tumor cells to demonstrate its requirement in maintaining growth/survival of these cells?

3. All the IPs were done with exogenously expressed proteins. Endogenous IP should be done with ST-EPN-YAP1 and ST-EPN-RELA tumors to confirm that (a) YAP1 pulls down more TEAD from ST-EPN-YAP1 tumors as we should expect based on Figure 1h and (b) YAP1-MAMLD1 but not YAP1 WT co-IP with endogenous NFIA and NFIB. According to Figure 7f, binding to NFIB seems to be mediated by YAP1 rather than MAMLD1, contrary to the authors claim.

4. The contribution of NFIA/B to YAP1-MAMLD1-mediated transcription/tumorigenesis is not clearly shown. What percentages of YAP1-MAMLD1-bound enhancer regions contain NFIA/B motifs? Among YAP1-MAMLD1/NFI co-bound regions, what percentages are positive for H3K27ac? A list of genes that are bound by YAP1-MAMLD1 and contain NFIA/B motifs along with H3K27ac status should also be added to the supplementary table. The author should also demonstrate actual NFI binding to select regions bound by YAP1-MAMLD1 by ChIP assay, and suppression of gene expression by Nfia-En. The authors mentioned in Discussion that NFIA/B are known to bind to canonical YAP target genes CTGF and CYR61. Western blot should be performed to show that the upregulation of these genes by YAP1-MAMLD1 (Fig 6c) is suppressed by with Nfia-En-GFP but not GFP control. The authors stated in Discussion that "Because of the substantial number of escaper cells in Nfia-En-electroporated animals (Figure 7P), however, these mice still developed Nfia-En (EGFP)-negative YAP1-MAMLD1-driven tumors". This data need to be included in the manuscript to demonstrate that Nfia-En truly suppress tumorigenesis. Also, shouldn't we expect to see survival differences between Nfia-En and EGFP (control) mice despite the escapers?

5. The canonical function of MAMLD1 in Notch signaling is not addressed. Does YAP1-MAMLD1 bind to classic MAMLD1-binding partners? Could they possibly be recruited by YAP1-MAMLD1 to regions bound by TEAD to promote transcription? Or does YAP1-MAMLD1 can also bind to and regulates classical Notch-regulated genes? These possibilities should be investigated and added to the manuscript.

Minor comments:

6. Figure 1f-g: Resolution is too low. Please replace with more clear images or IF analysis.

7. Figure 4e,f: why were only 513 genes were selected out of 1000 genes? The conservation between human and mouse should be much higher. Why weren't mouse EPN-RELA tumors included in the analysis? Do they group with human EPN-RELA tumors?

8. Figure S5d: why wasn't this shown in 3-d plot as figure 4t?

9. Figure 5a: The overall overlap between all YAP-bound peaks and H3K27ac peaks should also be presented. It is unclear why only group-specific enhancers are analyzed. Is it possible that enhancers shared by other subtypes could also involve in YAP1-MAMLD1-driven tumorigenesis? Does overall YAP bind to mostly enhancers in ST-EPN-YAP1 tumors as previously reported in other model systems? What about in EPN-RELA tumors? More detailed comparisons should be included in the supplementary figure.

10. Figure 5b: there appears to be little overlap in Yap binding among the three ST-EPN-YAP1 tumors. A supplementary graph should be provided to show the peak overlaps among all 6 six tumors (ST-EPN-YAP1 #1-3 and EPN-RELA #1-3) analyzed.

11. Figure 7p: this should be removed since it contains cells both transfected and untransfected cells. Instead, IF analysis should be done with luc to identify YAP1-MAMLD1-expressing cells.

Reviewer #3 (Remarks to the Author): In utero electroporation expert

In this work, Pajtler and colleagues demonstrate that the fusion of the Hippo pathway regulator YAP1 to the mastermind-like protein MAMLD1, an event frequently observed in supratentorial (ST) ependymomas (EPNs), is sufficient for its own nuclear translocation and the malignant transformation of cerebral neuronal stem cells. After 7 postnatal days, cells electroporated with the fusion plasmid were mostly located nearby the ventricular zone, suggesting a migration impairment, which was paralleled by incomplete differentiation and increased proliferation. Importantly, the translocation did not rely on the phosphorylation state of serine residue 127 (S127) of YAP1, but rather was mediated by the fusion partner. Cerebral precursors electroporated with plasmids encoding for YAP1 wild type form overexpression, unphosphorylated or truncated YAP1 failed to induce tumors in vivo in the long term. Notably, human ST-EPN-YAP1 tumors shares important molecular similarities with the mouse model employed. In addition, the observation that no tumor formation was observed in cells electroporated with YAP1-MAMLD1 fusion plasmid carrying a mutation at the interaction site with the transcriptional enhancer factor TEAD, confirmed that the fusion protein-induced tumorigenesis likely depends on alterations of transcriptional control. Finally, since animals electroporated with a plasmid encoding for YAP1 form carrying a nuclear localization signal (NLS)-conjugated domain failed to induced tumors, the authors investigated whether MAMLD1 activity was not only restricted to the nuclear translocation, but also concurred to the cell transformation process. The observed in vitro interaction between YAP1-MAMLD1 and Nuclear factor I (NFI) proteins and the reduced proliferation of cells transfected with YAP1-MAMLD1 observed after suppression of NFI targets confirmed the participation of MAMLD1 in the hyperproliferative process.

Minor comments

The article is a very interesting piece of work even for the wide readership of Nature Communications that may include non-experts. Indeed, the article is well written, with a clear introduction, well described result section and thoughtful discussion. The parallel between animal data and human data is for sure one of the strength on the whole work. The flow of experiments their logic and the design of control experiments is in my opinion strong.

I have however some issues that should be addressed:

- 1) The last Result's section ("Nuclear Factor I Proteins interact with MAMLD1 and are required for Fusion-Driven Hyperproliferation"), is a little more difficult to read and the reader may get a little lost. In this part, the authors should provide more information regarding the physiological role of Nfia and the rationale of the done experiment. It is quite difficult to understand why after the electroporation with the Nfia-En-IRES-EGFP construct authors inspect only the proliferation of cortical neural progenitors not considering other parameters. If there is non-stringent reason, authors should investigate also the other parameters.
- 2) One of the major strengths of the manuscript reside in the possibility of classification this type of tumour trough a thorough molecular characterization rather than clinical symptoms. This should be expanded in the discussion.
- 3) The description of the methods for in utero electroporation should be expanded and not only provided through a reference to another article.

Reviewer #1

An important paper describing a mouse model for YAP fusions in ependymoma. Not only do the authors generate a new model, they use this model and comparative biology with human tumors to gain insights into mechanisms of transformation by YAP fusion proteins, through activation of NFI and TEAD targets.

We thank the reviewer for their constructive comments on the manuscript. We therefore revised it extensively according to their suggestions.

My suggestions:

1. For nuclear localization of of p-YAP1 (S127), this is an important point, and IHC staining not so clear (and sometimes not so accurate with phospho antibodies). Can the phospho antibody also be analyzed in western localization experiments, in Fig 1 h and i. Also regarding this point, S128 phosphorylation augments nuclear localization of YAP, while S127 inhibits. Can the authors check the specificity of their S127 phospho antibody by checking whether this phosphorylation is gone in the S127A mutant, in Fig 2?

We fully agree with reviewer #1 that a Western Blot using an anti-p-YAP1 (S127) would strengthen our finding. We have therefore performed additional Western Blot analyses to test for phosphorylation of the S127 site. We analyzed the S127 phosphorylation status of YAP1 in primary human tumor samples of both ST-EPN-YAP1 and ST-EPN-RELA. We found a band at 140kDa that corresponds to the YAP1-MAMLD1 fusion protein phosphorylated at S127 in the nucleus of ST-EPN-YAP1 samples only. Phosphorylated wildtype YAP1 (p-YAP1) was clearly detected in the cytoplasmic compartment only. This newly generated data further corroborates findings from our IHC analysis and was added as Fig. 1j (page 4, lane 30-32). Notably, we found that p-YAP1 expression levels in ST-EPN-RELAs were low.

Reviewer #1 additionally raises an important point by stating that specificity of the pYAP1 (S127) antibody should be checked. We did accordingly and validated specificity of the p-YAP1 (S127) antibody. Using the p-YAP1 (S127) antibody, we performed immunostaining of E15.5 ventricular zone cells in cerebral cortices electroporated with *YAP1-MAMLD1-HA-IRES-GFP* and *YAP1 (S127A)-HA-IRES-GFP* at E13.5. While the fusion protein, which is phosphorylated at S127, was clearly observed with this antibody, no signal was detected in the YAP1 (S127A) mutant-expressing brain under the same IHC conditions. This indicates a high specificity of the p-YAP1 (S127) antibody that we used in this study. This additional data is now included as a separate supplementary figure (Supplementary Fig. 8) (page 12, lane 37).

2. In Fig 3, can neurite outgrowth and formation of cortical structures in control conditions be better delineated (perhaps in a supplemental figure) to better illustrate failure of these processes by fusion proteins?

We thank reviewer #1 for the important suggestion to better delineate phenotypical changes of cortical structures. To address this point, we performed additional immunostainings (P7) of brains that were electroporated at E13.5 with *T2TP* and either *EGFP* (upper panels) or *YAP1-MAMLD1-IRES-EGFP* using anti-GFP and Ki67 antibodies. While neurite outgrowth can be clearly shown in the controls, these morphological changes were absent in *YAP1-MAMLD1* electroporated cells. In contrast, *YAP1-MAMLD1* electroporated cells showed a high proliferation rate. According to the reviewer's suggestion, we added a supplementary figure to better illustrate the morphology of electroporated cells in the cerebral cortex at P7 (Supplementary Fig. 3a) (page 5, lane 38).

3. Page 6 and elsewhere, mice develop neurological signs not neurological symptoms (needs to be corrected in 7 places), since mice can't communicate how they feel, minor issue.

This was adjusted accordingly. We replaced “symptoms” with “signs” throughout the manuscript.

4. Lettering in Fig 4 is wrong in text, PCA analysis in text refers to IF staining.

We carefully corrected the mistakes.

5. H and E section in Fig S5 of poor quality, can this be replaced with a figure that is more cleanly stained and photographed?

Staining was repeated for the revision and the previous figure was replaced. In addition, the figure was replaced to better show staining of the tumor cells (Supplementary Fig. 5d).

Reviewer #2 (Remarks to the Author): Yap expert

The authors demonstrate the oncogenic role of a novel YAP1-MAMLD1 fusion protein in supratentorial ependymoma. The YAP1-MAMLD1 fusion protein preferentially localizes to nucleus, which is driven by MAMLD1 independent of S127 phosphorylation. In addition, MAMLD1 facilitate the binding of YAP1-MAMMLD1 fusion protein to NFI. The authors further demonstrate that YAP1, MAMLD1, and NFI are all required for the oncogenic transformation using an electron transportation model. Overall, the experiments are well designed, and properly controlled. The findings are interesting. However, some additional experiments will help strengthen the overall impacts of the manuscript.

We thank the reviewer for their overall positive judgement about the impact of our manuscript.

Major comments:

1. Data in Figure 1a-c are largely re-plotting of the same data as what was published by the group in Cancer Cell 2015 except for five new samples. Figure 1c is nearly identical to Figure 2c in Cancer Cell 2015 paper. This raises potential copyright infringement issue and should be removed from the current manuscript.

We appreciate reviewer #2's comment on this issue. Fig. 1b were removed from the original manuscript. Since ST-EPN-YAP1 tumors are quite rare, we could only add five new samples to the initially submitted version of the manuscript. Given reviewer #2's concerns we have made substantial efforts to increase the ST-EPN-YAP1 cohort. This resulted in the largest existing DNA methylation dataset of ST-EPN-YAP1 tumors (n = 45) (page 4, lanes 3-8). The unsupervised clustering analyses were repeated using the ST-EPN-RELA dataset as control. From the resulting TSNE plot we could still appreciate two independent and clearly separated cohorts relating to ST-EPN-YAP1 and ST-EPN-RELA. Fig. 1a was replaced by the new clustering accordingly.

2. Figure 1h: It is important to assess the relative expression levels of YAP1-MAMLD1 to YAP1 WT in ST-EPN-YAP1 tumors, and to YAP1 WT in ST-EPN-RELA tumors. Please repeat the western to show YAP-fusion and WT protein detection on the same blot. A p-YAP S127 blot need also be done with the same samples to demonstrate that phosphorylation-independent nuclear translocation in ST-EPN-YAP1 tumors. This is very important because all the in vivo experiments were done with exogenously expressed YAP1-MAMLD1.

We thank reviewer #2 for this comment. For Fig. 1h, we performed detection of YAP1 fusion and WT protein on the same blot. The uncropped Western Blot is now shown in Supplementary Fig.9. We also examined subcellular localization of phospho-YAP1-MAMLD1 in human primary EPNs. As described in response to reviewer 1, additional western blot analyses to test for phosphorylation of the S127 site were performed. We analyzed the S127 phosphorylation status of YAP1 in primary human tumor samples of both ST-EPN-YAP1 and ST-EPN-RELA. We found a band at 140kDa in both the cytoplasm and the nucleus of ST-EPN-YAP1 only that corresponds to

the YAP1-MAMLD1 fusion protein phosphorylated at S127. Phosphorylated wildtype YAP1 (p-YAP1, 70kDa) was restricted to the cytoplasmic compartment. This newly generated data indicates phosphorylation-independent nuclear translocation of the YAP1-MAMLD1 fusion protein in ST-EPN-YAP1 tumors. We added a new figure demonstrating these findings (Fig. 1j) (page 4, lanes 30-32).

Also, contrary to the authors' claim in page 7, the overall mRNA levels of YAP (both fusion and WT) is significantly lower comparing to the rest of the subtypes as shown in Figure S1. It seems that YAP1 WT proteins are expressed at comparable levels between ST-EPN-YAP1 and ST-EPN-RELA tumors. If endogenous YAP1-MAMLD1 is very lowly expressed, could it really be the oncogenic driver? Could knockdown of the YAP1-MAMLD1 fusion protein be done with primary ST-EPN-YAP1 tumor cells to demonstrate its requirement in maintaining growth/survival of these cells?

The reviewer raises an important point here. In the original version of the manuscript, we are stating that the overall YAP1 mRNA expression level in human ST-EPN-YAP1 tumors was not higher than in other intracranial molecular ependymoma groups. We agree with reviewer #2 that the YAP1 expression level seems indeed relatively low in ST-EPN-YAP1 compared to other subgroups (Supplementary Fig. 1c). However, the absolute level of YAP1 mRNA within these tumors is quite high. To depict this clearly, we added a new figure showing relative YAP1 mRNA level to those of the other genes in a representative ST-EPN-YAP1 tumor as a Supplementary Fig. 1d (page 4, lanes 24-25). Please note that its expression is higher than the mean combined with the standard deviation across all genes expressed in a ST-EPN-YAP1 tumor. In addition, it is of high importance that the resulting fusion protein is spontaneously translocated to the nucleus which according to our model exerts its oncogenic activity through direct functional interactions with other partners, e.g. TEAD and NF1A, which might not be dependent on high

expression levels.

While we agree regarding the importance of knockdown experiments in primary tumor cells, please be aware that to date unfortunately no cell lines or other models of ST-EPN-YAP1 tumors have been established. In fact, this study is the first reporting a ST-EPN-YAP1 model. Since short-term cultures of primary tumors could not be established so far due to unavailability of primary ST-EPN-YAP1 cells, we are afraid that knockdown experiments in primary cells cannot be performed at this stage.

3. All the IPs were done with exogenously expressed proteins. Endogenous IP should be done with ST-

EPN-YAP1 and ST-EPN-RELA tumors to confirm that (a) YAP1 pulls down more TEAD from ST-EPN-YAP1 tumors as we should expect based on Figure 1h and (b) YAP1-MAMLD1 but not YAP1 WT co-IP with endogenous NFIA and NFIB. According to Figure 7f, binding to NFIB seems to be mediated by YAP1 rather than MAMLD1, contrary to the authors claim.

We apologize for the confusion created by Figure 7f in the initially submitted manuscript. To make a rigid conclusion, we further added proper negative controls. Please be aware that all IP results were replaced with new ones (Fig. 7a, b).

Although we fully agree with reviewer #2 that endogenous IP would be the best option, we could not perform this multiple times due to limited amounts of human primary tumors. Nevertheless, as requested by reviewer #2 we could manage to perform additional IPs with primary material and saw a pull down of YAP1-MAMLD1 but not YAP1 WT with endogenous NFIB in one representative primary tumor. This is now included as a new supplementary figure (Supplementary Fig. 7a) (page 9, lanes 10-12). The potential interaction between YAP1-MAMLD1 and NFIA protein was also supported by ChIP-seq analysis showing that 96% of YAP1 binding loci in YAP1-MAMLD1 is shared with NFIA. More details about this observation are also provided below. These data strongly supported our hypothesis that YAP1-MAMLD1 interacts with NFIA/B protein in human primary ST-EPN-YAP1 tumors.

4. The contribution of NFIA/B to YAP1-MAMLD1-mediated transcription/tumorigenesis is not clearly shown. What percentages of YAP1-MAMLD1-bound enhancer regions contain NFIA/B motifs? Among YAP1-MAMLD1/NFI co-bound regions, what percentages are positive for H3K27ac? A list of genes that

are bound by YAP1-MAMLD1 and contain NFIA/B motifs along with H3K27ac status should also be added to the supplementary table.

We are thankful to the reviewer #2 for providing the important suggestions to further investigate and verify the NFIA/B activity in collaboration with YAP1. From an *in silico* analysis of YAP1 group-specific peaks, it was possible to observe strong enrichment of NFI since the motifs of this transcription factor were detected to be present in 87% of peaks. To further confirm this effect *in vitro*, we performed additional NFIA ChIP sequencing analyses on the identical three YAP1 tumors, which were used for YAP1 ChIP-seq in this study. The general QC summary of this dataset is included into the corresponding supplementary table “ChIP-seq QC”. In order to investigate the group specificity of NFIA peaks we used the same approach by focusing on filtered YAP1 peaks based on the exclusion of overlap with common peaks from the ST-EPN-RELA group. We found that 96% (1201 out of 1246) of ST-EPN-YAP1 group specific peaks were overlapping with NFIA peaks. Notably, for ST-EPN-RELA specific YAP1 peaks the overlap was only 13% (337 out of 2464). Further focusing on peaks overlapping with group specific enhancers we confirmed that the pattern remained the same: ST-EPN-YAP1 – 97.5% (278 out of 285), ST-EPN-RELA – 11% (121 out of 1028). These additional ChIP-seq results are now referred to in the text and are included as additional figure panels (Fig. 7c and Supplementary Fig. 7b, c) in the revised manuscript (page 9, lanes 12-16). As suggested by reviewer #2, the status of YAP1 peaks overlapping with NFIA peaks was included as an additional column into the Supplementary table 1 “TAD genes ST-

EPN-YAP1”.

The author should also demonstrate actual NFI binding to select regions bound by YAP1-MAMLD1 by ChIP assay, and suppression of gene expression by Nfia-En. The authors mentioned in Discussion that NFIA/B are known to bind to canonical YAP target genes CTGF and CYR61. Western blot should be performed to show that the upregulation of these genes by YAP1-MAMLD1 (Fig 6c) is suppressed by with Nfia-En-GFP but not GFP control.

As mentioned above, a lack of human cell lines and representative models of ST-EPN-YAP1 prevented us from testing suppression of YAP1-MAMLD1 target genes by Nfia-En. However, as suggested by reviewer #2, we focused on canonical YAP1 targets *CTGF* and *CYR61*, since these genes are activated more strongly by YAP1-MAMLD1 than YAP1 wildtype (Fig. 6a, c) and were bound by both YAP1 and NFIA in human ST-EPN-YAP1s (Supplementary Fig. 7b, c) (page 9, lanes 14-16). We tested if Nfia-En blocked activation of *CTGF* and *CYR61* caused by exogenous YAP1-MAMLD1 using HEK293T cells. Indeed, overexpression of Nfia-En strongly repressed expression of *CTGF* and *CYR61*. These supporting results are now included as Supplementary Fig. 7h (page 9, lanes 23-27).

The authors stated in Discussion that “Because of the substantial number of escaper cells in Nfia-En-electroporated animals (Figure 7P), however, these mice still developed Nfia-En (EGFP)-negative YAP1-MAMLD1-driven tumors”. This data need to be included in the manuscript to demonstrate that Nfia-En truly suppress tumorigenesis. Also, shouldn’t we expect to see survival differences between Nfia-En and EGFP (control) mice despite the escapers?

Due to the nature of the experiments, we cannot estimate how many cells were transfected in individual animals by electroporation. This causes difficulty in comparing control GFP with Nfia-En GFP conditions in terms of tumor latency (median survival: 33days for EGFP alone and 32.5 days for Nfia-En-EGFP). Therefore, to examine whether Nfia-En-expressing cells prevent tumor formation, we examined GFP+ cells in tumors. As shown in the new Fig. 7o-q and Supplementary Fig. 7q-t, we only observed a few GFP+ cells in Nfia-En-expressing tumors. Meanwhile, most of the control tumor cells expressed Mock GFP (page 9, lanes 33-38). Together with the data that significantly higher percentages of the escapers were observed in P0 brains electroporated with Nfia-En than in controls (see Fig. 7n), the tumors from mice electroporated with Nfia-En most likely consisted of escapers.

5. The canonical function of MAMLD1 in Notch signaling is not addressed. Does YAP1-MAMLD1 bind to classic MAMLD1-binding partners? Could they possibly be recruited by YAP1-MAMLD1 to regions bound by TEAD to promote transcription? Or does YAP1-MAMLD1 can also bind to and regulates classical Notch-regulated genes? These possibilities should be investigated and added to the manuscript.

According to the points raised by reviewer #2 we performed an additional analysis. As a result, we did not observe specific activation of Notch effectors in ST-EPN-YAP1s when focusing on the GO analysis of the enhancer-associated genes in connection to YAP1 (see new GO EAG ST-EPN-YAP1 in Supplementary Table 1). Furthermore, we did not see enrichment of Notch effector binding sites in YAP1 peaks of ST-EPN-YAP1s in the table.

Since MAMLD1 has been known to activate the Hes3 promoter, we generated YAP1-MAMLD1 with a L103P mutation, which inhibits MAMLD1 transactivation for the Hes3 promoter (Fig. 3B in Fukami et al., 2008, see right panel) as mentioned in the DISCUSSION section. Given that this construct did not prevent YAP1-fusion-driven tumor formation, we conclude that MAMLD1-mediated Hes3 activation is not relevant for tumor formation in this context.

Regarding classic MAMLD1-binding partners, to our best knowledge, no classic MAMLD1-binding partners have been reported.

Minor comments:

6. Figure 1f-g: Resolution is too low. Please replace with more clear images or IF analysis.

We replaced these pictures accordingly with enlarged high-resolution ones.

7. Figure 4e,f: why were only 513 genes were selected out of 1000 genes? The conservation between human and mouse should be much higher. Why weren't mouse EPN-RELA tumors included in the analysis? Do they group with human EPN-RELA tumors?

We appreciate the comments made by reviewer #2 pointing out missing aspects in the investigation of the mouse model. The orthologous genes were strictly selected by additional filtering effect on the number of genes from the combination of Affymetrix probes representing the same gene. The top 1000 most differentially expressed gene probes covered 671 genes. Among the 671 genes, 531 genes were human-mouse orthologs. To make this process clearer, the selection procedure was described in more detail in the Methods (page 15, lanes 12-19).

We also thank reviewer #2's very important questions regarding inclusion of human and mouse RELA tumors in cross-species comparison. In order to answer to these questions, we analyzed the transcriptome dataset of human and mouse RELA tumors and presented principal component analyses between human and mouse tumors in Supplementary Fig. 4e (page 7, lanes 11-14 and page 15, lanes 20-27). To control for a major species impact, we selected 308 orthologs from YAP1-RELA differentially expressed genes that are shared between human tumors and models (Supplementary Table 1 "YAP1-RELA DEGs mice human") for the analysis. Please also note that these tumor data were combined with transcriptome data of pediatric glioblastoma multiforme (pGBMs) to verify the tumor specificity. PCA method was applied and demonstrated that YAP1 models resemble primary human ST-EPN-YAP1

tumors. The corresponding Supplementary Fig. 4e in the manuscript was revised and updated accordingly.

8. Figure S5d: why wasn't this shown in 3-d plot as figure 4t?

This was done to highlight the effects of principal components. In order to avoid confusion and maintain the same format after inclusion of RELA mice model into the comparison, the corresponding Supplemental Fig. 4e was updated and also provided as 3D plot.

9. Figure 5a: The overall overlap between all YAP-bound peaks and H3K27ac peaks should also be presented. It is unclear why only group-specific enhancers are analyzed. Is it possible that enhancers shared by other subtypes could also involve in YAP1-MAMLD1-driven tumorigenesis? Does overall YAP

bind to mostly enhancers in ST-EPN-YAP1 tumors as previously reported in other model systems? What about in EPN-RELA tumors? More detailed comparisons should be included in the supplementary figure.

The group-specific enhancers were initially selected for the comparison with YAP1 peaks in order to allow the integration of enhancer-associated genes that were previously detected in the ependymoma enhancer landscape study (Mack et al, 2018). The details of these genes' connection to YAP1 activity are stated in the supplementary tables "TAD genes ST-EPN-YAP1" and "TAD genes ST-EPN-RELA". However, to further cover the aspect of full enhancer landscapes as suggested by reviewer #2 we also checked the overlap with all enhancers including those shared between subgroups (n=45495). We observed that 87% of YAP1 peaks (1084 out of 1246) specific for ST-EPN-YAP1 were overlapping with any ependymoma enhancer, while for the ST-EPN-RELA the proportion was almost 50% lower – 44.6% (1100 out of 2464). We have included this information as Supplementary Fig. 5c (page 7, lanes 28-31, 44 and page 8, lanes 1-2).

10. Figure 5b: there appears to be little overlap in Yap binding among the three ST-EPN-YAP1 tumors. A supplementary graph should be provided to show the peak overlaps among all 6 six tumors (ST-EPN-YAP1 #1-3 and EPN-RELA #1-3) analyzed.

Indeed, there is variance observed between the YAP1 ChIP-seq datasets reflected by the number of confident peaks per sample. For better illustration we created an additional supplementary figure that gives a more detailed overview of peak overlaps among all samples (Supplementary Fig. 5d). From this

figure it can be appreciated that the largest number of common YAP1 peaks is observed across all samples combined (n=6) as we already demonstrated in Supplementary Fig. 5b. Nevertheless, group specific peaks remain in the top10 common overlapping peak sets both for ST-EPN-RELA and ST-EPN-YAP1 despite the differences in total number of confident peaks per sample. We marked them accordingly in the new figure.

11. Figure 7p: this should be removed since it contains cells both transfected and untransfected cells. Instead, IF analysis should be done with luc to identify YAP1-MAMLD1-expressing cells.

In the initially submitted manuscript, we used YAP1 as a marker for successfully electroporated cells. As YAP1 is a nuclear protein, we can quantify labeled cells more convincingly than using Luc staining. To show that non-electroporated cells do not express YAP1 in the cortical plate at P0, we now added IHC pictures of electroporated whole brains in Supplementary Fig. 7i-l.

Reviewer #3 (Remarks to the Author): In utero electroporation expert

In this work, Pajtler and colleagues demonstrate that the fusion of the Hippo pathway regulator YAP1 to the mastermind-like protein MAMLD1, an event frequently observed in supratentorial (ST) ependymomas (EPNs), is sufficient for its own nuclear translocation and the malignant transformation of cerebral neuronal stem cells. After 7 postnatal days, cells electroporated with the fusion plasmid were mostly located nearby the ventricular zone, suggesting a migration impairment, which was paralleled by incomplete differentiation and increased proliferation. Importantly, the translocation did not rely on the phosphorylation state of serine residue 127 (S127) of YAP1, but rather was mediated by the fusion partner. Cerebral precursors electroporated with plasmids encoding for YAP1 wild type form overexpression, unphosphorylated or truncated YAP1 failed to induce tumors in vivo in the long term. Notably, human ST-EPN-YAP1 tumors shares important molecular similarities with the mouse model employed. In addition, the observation that no tumor formation was observed in cells electroporated with YAP1-MAMLD1 fusion plasmid carrying a mutation at the interaction site with the transcriptional enhancer factor TEAD, confirmed that the fusion protein-induced tumorigenesis likely depends on alterations of transcriptional control. Finally, since animals electroporated with a plasmid encoding for YAP1 form carrying a nuclear localization signal (NLS)-conjugated domain failed to induced tumors, the authors investigated whether MAMLD1 activity was not only restricted to the nuclear translocation, but also concurred to the cell transformation process. The observed in vitro interaction between YAP1-MAMLD1 and Nuclear factor I (NFI) proteins and the reduced proliferation of cells transfected with YAP1-MAMLD1 observed after suppression of NFI targets confirmed the participation of MAMLD1 in the hyperproliferative process.

Minor comments:

The article is a very interesting piece of work even for the wide readership of Nature Communications that may include non-experts. Indeed, the article is well written, with a clear introduction, well described result section and thoughtful discussion. The parallel between animal data and human data is for sure one of the strength on the whole work. The flow of experiments their logic and the design of control experiments is in my opinion strong.

We thank the reviewer #3 who recognizes the importance and novelty of our work.

I have however some issues that should be addressed:

1) The last Result's section ("Nuclear Factor I Proteins interact with MAMLD1 and are required for Fusion-Driven Hyperproliferation"), is a little more difficult to read and the reader may get a little lost. In this part, the authors should provide more information regarding the physiological role of Nfia and the rationale of the done experiment. It is quite difficult to understand why after the electroporation with the Nfia-En-IRES-EGFP construct authors inspect only the proliferation of cortical neural progenitors not considering other parameters. If there is non-stringent reason, authors should investigate also the other parameters.

We thank reviewer #3 for the valuable comment. We extensively revised the last section and included new figures to better explain the rationale behind our experiments and guide the reader. As suggested by reviewer #3, we examined apoptotic cell death of electroporated cells as additional parameter using an anti-cleaved caspase-3 antibody (Fig. 7m and Supplementary Fig. 7m-p) (page 9, lanes 32-33).

2) One of the major strengths of the manuscript reside in the possibility of classification this type of tumour trough a thorough molecular characterization rather than clinical symptoms. This should be expanded in the discussion.

We fully agree with reviewer #3. We expanded in the first paragraph of the DISCUSSION accordingly and included an additional paragraph stating that precise molecular classification represents an important precondition for any potential future stratification that may include careful therapy de-escalation in ST-EPN-YAP1 patients but will also help to make use of subgroup specific therapeutic targets (page 10, lanes 2-9).

3) The description of the methods for in utero electroporation should be expanded and not only provided through a reference to another article.

We added more information about the methods for in utero electroporation in the METHODS (page 14, lanes 34-38).

REVIEWERS' COMMENTS:

Reviewer #1 (Remarks to the Author):

revised manuscript adequately addresses points raised in prior review

Reviewer #2 (Remarks to the Author):

The authors did a thorough job addressing my concerns. I have no further comments. Recommend for publication.

Reviewer #3 (Remarks to the Author):

The authors have done a good job at revising the manuscript by providing the rationale for investigating the role of Nuclear Factor I Proteins as YAP1-MAMLD1 cofactors involved in its oncogenic activity (page 9). Importantly, the fact that, when compared to control, Nfia-En-EGFP overexpressing cells show similar levels of apoptotic cell death (Supplementary Fig. 7m-p, page 9) but less proliferation, reinforces their involvement in YAP1-MAMLD1-driven hyperproliferation of cortical neural progenitors. Supplementary Fig. 7m-p is a valuable addition given the ability of YAP1 to bind p73 after nuclear translocation, with the consequent activation of the cell death pathway.

As suggested, the importance of the molecular classification of this type of tumors and its possible impact in individuating specific therapeutic targets has been highlighted in the first paragraph of the discussion (page 10). This new version makes this work more interesting for the broader readership.

Sufficient details have been provided for the in utero electroporation technique in the 'Methods' section (page 14).

I support the publication of the article and I have no further questions.